# Unmasking the perching effect of the pioneer Mediterranean dwarf palm *Chamaerops humilis* L.

**Víctor González-García** [1]ᴼ*, **Pedro J. Garrote**[2]ᴼ, **Jose M. Fedriani** [3,4]ᴼ*

**1** Atlantic Botanical Gardens, Gijon, Asturias, Spain, **2** Centre for Applied Ecology "Prof. Baeta Neves" (CEABN/InBIO), Institute Superior of Agronomy, University of Lisbon, Tapada da Ajuda, Lisbon, Portugal, **3** Desertification Research Centre CIDE, CSIC UV GV, Moncada, Valencia, Spain, **4** Doñana Biological Station (EBD—CSIC), Seville, Spain

ᴼ These authors contributed equally to this work.
* victor.gon.gar@hotmail.com (VGG); fedriani@csic.es (JMF)

**Data Availability Statement:** All data files are fully available at: https://figshare.com/articles/dataset/PerchigEffectDwarfPalm_xlsx/19642446.

**Funding:** This work was supported by the Portuguese Foundation for Science and

## Abstract

Although farmlands are the most extensive terrestrial biomes, the abandonment of traditional agriculture in many parts of the world has brought opportunities and challenges for the restoration of such human-disturbed habitats. Seed arrival is a crucial necessary ecological process during plant recolonization that can be enhanced by the use of the so-called "perch plants". Little is known, however, about whether the seed arrival via frugivorous birds is affected by the spatial distribution of the perch plants in disturbed habitats. To evaluate several spatial aspects of "perching" effect, we used a spatially explicit approach in two disturbed plots within the Doñana National Park (SW Spain). Specifically, we chose as study system the pioneer Mediterranean dwarf palm *Chamaerops humilis* L., which is often used as a perch by a variety of frugivorous bird species. A total of 289 *C. humilis* individuals were sampled in search of bird feces (N = 2998) and dispersed seeds (N = 529). Recorded seeds belonged to six different woody species from five different families. Nine bird species from six different families were recorded using *C. humilis* as perches. GLMs analyses indicated that taller *C. humilis* males with higher numbers of spatially associated woody species received more dispersed seeds. We detected a random spatial structure of bird feces and dispersed seeds in one study plot, while a nonrandom spatial structure was found in the other one, where isolated *C. humilis* received a higher number of bird feces and dispersed seeds than expected under spatial null models. The difference in spatial patterns between both study plots could relate, among other factors, to their different state of development in the ecological succession. Most of dispersed seeds were concentrated in a small number of *C. humilis* individuals, usually male and large ones, that acted as "hotspots" of seed arrival. The fact that frugivorous birds in one study site visited most often isolated *C. humilis* questions the aggregated spatial structure of revegetation designs typically used in restoration projects. This study reveals novel spatial aspects of the "perching" effect which could be helpful in the restoration of human-disturbed habitats worldwide.

Technology to PJG (SFRH/BD/130527/2017) and
by a grant of the Spanish Ministry of Education and
Science to JMF (PGC2018-094808-B-I00). Logistic
and technical support was provided by ICTS-RBD.
The authors acknowledge support of the
publication fee by the CSIC Open Access
Publication Support Initiative through its Unit of
Information Resources for Research (URICI). The
funders had no role in study design, data collection
and analysis, decision to publish, or preparation of
the manuscript.

**Competing interests:** The authors have declared
that no competing interests exist.

## Introduction

Croplands and grazing lands have become the largest terrestrial biomes, occupying almost the 40% of Earth's land surface [1, 2]. However, in the Mediterranean Europe, areas destined to agriculture have been reduced by 34.2% in the last half century. In Spain, specifically, they have decreased by 54.8% [3, 4]. This abandonment of farmlands represents an opportunity and a challenge to boost biodiversity through natural (and facilitated) revegetation and reforestation [5], and thus to restore habitats to the stages prior to human disturbances. Restoration projects are generally expensive given that they have been usually focused on planting dominant tree species [6]. Although many of such restoration projects have been successfully [7], they could be unaffordable when the affected area is too large, limiting the full recovery of vegetation structure [8]. Natural regeneration acts more widely and is less expensive than tree plantations [5], although its intensity varies by region and habitat. In arid and semiarid areas with low primary productivity such as the Mediterranean basin, natural regeneration is clearly slower [7, 9] than in more humid and productive environments like tropics or temperate regions, since the water availability for plants is an evident limitation [10]. Some other constrains in human-disturbed habitat restoration are inter- and intraspecific competition from herbaceous plants [11], the border effect or the past landscape pattern [12], and the limited propagule pressure and dispersal capacity of seeds [9, 13, 14].

Seed dispersal plays a key role in habitat dynamics and structure of plant communities [15]. Seed dispersal facilitates plant colonization of vacant areas and thus, the recovery of vacant human-altered habitats [16]. Birds and mammals are the main seed dispersers of most fleshy-fruited plants [17]. Many frugivorous birds often use tree and shrub branches as "perches", i.e. sites where they defecate and regurgitate large number of viable seeds [18–20]. Repeated visits to the same perches locally increase seed arrival and thus can trigger nucleation processes [20]. Human-disturbed habitats are more prone to these nucleation effects, since they are often vacant habitats where perches are scarce, which promotes a marked seed clumping [21, 22]. However, we are not aware of any study that has quantified the patterns of seed clumping under perches using a spatially explicit approach.

Interestingly, in some cases, the same shrub species acting as perches also act as nurse plants, i.e. facilitating the emergence, growth and survival of other plant species, [23] which are designated as "beneficiary species", promoting the natural (re)colonization [24, 25]. *Chamaerops humilis* L., is a pioneer Mediterranean palm crucial in habitat recovery [26]. Our unpublished data indicate that these palms potentially act as perches for many frugivorous birds, which play important roles as seed dispersers and agents of the natural recolonization of human-disturbed areas by woody plants (e.g. [27]). The facilitative role of this keystone palm has been recently evaluated [28]. In particular, [4] the dwarf palm has been found positively associated with multiple woody plant species in disturbed vacant habitats. By contrast, the "perching" effect of *C. humilis* and its potential implications for the ecological restoration of human-disturbed landscapes remain unknown. In addition, very little is known about the pervasiveness of the natural "perching" effect and its spatial aspects and scales. On this matter, Spatial Point Pattern Analysis (SPPA) constitutes a powerful tool to evaluate this sort of questions [29].

In this study we evaluate spatial patterns on the role of *C. humilis* as perch for frugivorous birds dispersing woody plant seeds in human-disturbed habitats. To achieve our goal, we selected two study plots within Doñana National Park (SW Spain) that differ in their history of human-driven perturbations and vegetation. We used a spatially explicit approach to answer the following questions: (*i*) How pervasive is the "perching" effect in our two studied human-disturbed habitats? (*ii*) Are there differences between the two study plots in the role of *C.*

*humilis* as perch? And, if so, what are the factors underlying such differences? (*iii*) Do aggregated *C. humilis* receive more bird feces containing dispersed seeds? (*iv*) Do particular *C. humilis* individuals act as "hot spots" of seed arrival, and if so, why? And (*v*) Do the species of dispersed seeds correspond to the woody beneficiary species found spatially associated with the *C. humilis*?

## Material and methods

### Study species

*Chamaerops humilis* L. (Arecaceae), commonly known as "European fan palm" or "Mediterranean dwarf palm", is an evergreen shrub endemic to the western Mediterranean [30, 31]. This palm can grow up to 10m tall, although individuals are not usually taller than 2m [32]. Leaves are big, fan-shaped and displayed in numerous segments, exhibiting several long thorns on the petiole. It is dioecious, with fruits only growing on female plants. Flowering occurs in spring, from March to May. Spread takes place through seeds (sexually) or, less commonly, through division of adult trees (asexually) [30]. Seeds are mostly dispersed by medium-sized carnivores such as Eurasian badgers *Meles meles* and red foxes *Vulpes vulpes* [33] but also by ungulates (e.g. red deer *Cervus elaphus* and wild boar *Sus scrofa*) some of which occasionally disperse seeds by regurgitation [34, 35].

The facilitative role of *C. humilis* and its capacity of recolonizing disturbed habitats have made it widely used in ecological restorations [36], although its role as nurse plant has only recently been demonstrated [4]. *Chamaerops humilis* protects seeds and seedlings of woody species from high temperatures and drought, as well as from herbivory thanks to the spikes present in its leaves [4, 37]. Frequent visits of some seed dispersers such as red foxes to fruiting *C. humilis* could increase perceived predation risk for several species of rodents (e.g. *Apodemus sylvaticus*) and lagomorphs (e.g. *Oryctolagus cuniculus*) [35] and thus locally lessen small mammal seed and seedling predation of beneficiary plants.

In its typical habitat, the Mediterranean scrubland, *C. humilis* is often spatially associated with *Asparagus* L. spp., *Daphne gnidium* L., *Olea europaea* L. var. *sylvestris* (Mill.) Hegi, *Pistacia lentiscus* L., *Pyrus bourgaeana* Decne, and other woody species. These spatial associations are thought to be due to a combination of "nursing" and "perching" effects [4, 30], although the "perching" effect has not been tested yet.

### Study area and plots

The study was carried out within the Doñana National Park (SW Spain, on the west of the Guadalquivir River estuary). All permits necessary were granted by the National Park Service (ref. 2019/10) and the Junta de Andalucía (ref. 2019107300002261/IRM/MDCG/mes). The National Park presents a Mediterranean sub-humid climate, with most of rainfalls taking place from November to April, showing an average annual rainfall of 500-600mm [33], with hot dry summers (June-September) and mild humid winters (October-January).

We selected one early- and one late-successional study plots 10 km apart (Fig 1). Study areas are different in terms of vegetation, successional stage and human-use history [4, 38]. The early-successional plot (Matasgordas; 13.9 ha) has been historically and intensively affected by anthropic management of the vegetation and livestock pressure [39]. The eastern portion of this study plot was drastically modified in 1970 when most of woody plants (both shrubs and trees) were removed in order to create a "dehesa", becoming an open area for cattle grazing with scattered trees, mostly *Quercus suber* L. and *O. europaea* var. *sylvestris* [4, 16]. In 1996, the land became owned by Government and happened to be under the protection of the National Park boundaries, putting an end to the cattle grazing. Once livestock was removed

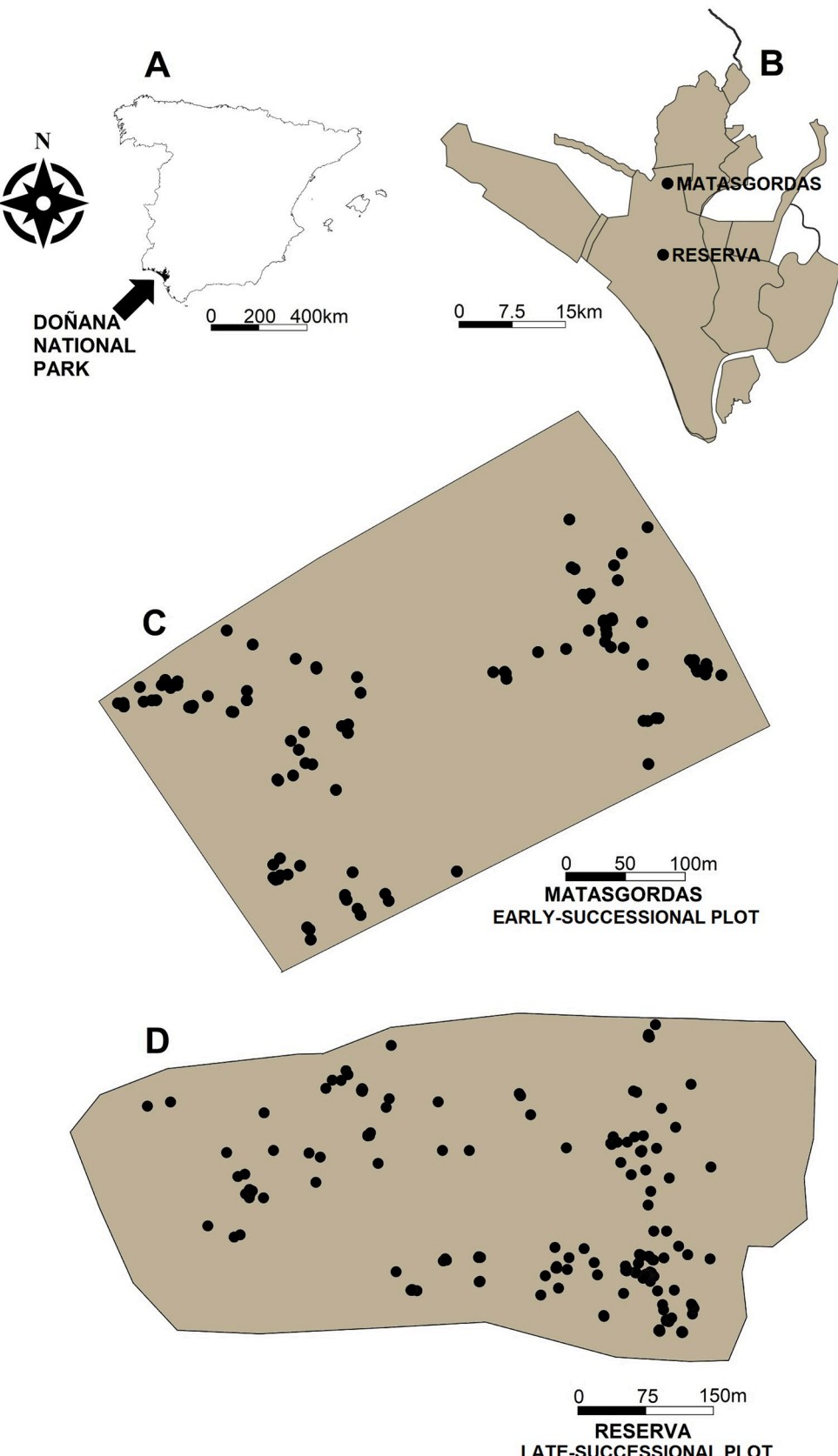

**Fig 1. The study plots.** (A) Darkened location of the Doñana National Park within (NW) Spain. (B) Doñana Natural Park with the two study plots highlighted in black (C) early-successional (Matasgordas) and (D) late-successional (Reserva) study plots located within the Doñana National Park. Every occurring *C. humilis* was georeferenced (black solid points).

from the area, native fleshy-fruited plants such as *C. humilis*, *D. gnidum*, *P. lentiscus* and *P. bourgaeana* started to recolonize their habitat via seed dispersal by birds and mammals [4, 16]. Due to such past human activity, the area is now formed, mainly, by two habitats: (*i*) Scrubland dominated by *P. lentiscus* shrubs with some *Q. suber* and *O. europaea* var. *sylvestris trees* [4, 33] and (*ii*) an old-field, where our study plot was set, which woody vegetation is mainly composed by animal-dispersed native plants such as *C. humilis*, *P. lentiscus*, *D. gnidium* and *P. bourgaeana*, *Asparagus aphyllus* L., *Halimium halimifolium* (L.) Willk and *Cistus salviifolius* L. and scattered *Q. suber* and *O. europaea* var. *sylvestris* trees [4]. The distribution of *C. humilis* in this plot is very aggregated, and best described by a double-clustered component process with a random component process (with 22% of isolated *C. humilis* individuals; see S1 File and S1 Table).

Doñana Biological Reserve is located in the centre of the National Park. It is laid on a coastal plain distinguished for its flatness and being separated from the sea by lagoons and mobile dune systems. However, it is this closeness to the sea which allows a humid and mild climate [40] and enables the existence of three different types of habitats: marshland, dunes and scrubland; being the ecotone between the scrubland and the marshes (locally known as "La Vera") where we set our late-successional study plot. This area has been historically managed by human for agriculture, hunting, cattle grazing and tree felling, especially *O. europaea* var. *sylvestris* and *Q. suber* [41]. Reserva was protected earlier than Matasgordas, in 1964 and it has been recovering ever since, being recolonised by animal-dispersed plants such as *A. aphyllus*, *C. humilis*, *Phillyrea angustifolia* L. or *R. ulmifolius* [4]. This scrubland is dominated by *H. halimifolium* and *Stauracanthus genistoides* (Brot.) Samp. with scattered trees of *Q. suber*, *O. europaea* var. *sylvestris* and *Pinus pinea* [33]. Our later-successional study plot presents higher density of shrubs than the one at the early successional study plot. Such a difference in shrub cover led to a greater abundance of fleshy fruits in the early-successional study plot, which likely makes this site more attractive to frugivorous birds [4]. The distribution of *C. humilis* in this plot was also best described by a double-clustered component process with a random component process (with 9% of isolated individuals; see S1 File and S1 Table).

## Data collection

Every target *C. humilis* (109 and 180 in the early and late successional study plots, respectively) was individually georeferenced with a submetric GPS Leica 1200. Mean separation distances were 219 and 223 m in the early and late successional study plots, respectively. The density of *C. humilis* plants in the late-successional plot (7.8 individuals / ha) was pretty similar to the late-successional plot (8.41 individuals / ha). Each study plot was visited eight and five times along the seed dispersal seasons (i.e. from September to November) of 2019 and 2020, respectively. To control for potential temporal variations on bird abundance and seed dispersal along the dispersal seasons, both study plots were sampled during the same sampling days. Rainy days were avoided because of the low bird activity and the feces wash out on *C. humilis*.

During the sampling, which lasted about 3 h per day in the early-successional plot and about 2 h in the late-successional plot, any observed bird perching on focal *C. humilis* individuals was recorded. Overall, nine bird species were recorded perching: *Cisticola juncidis*, *Lanius*

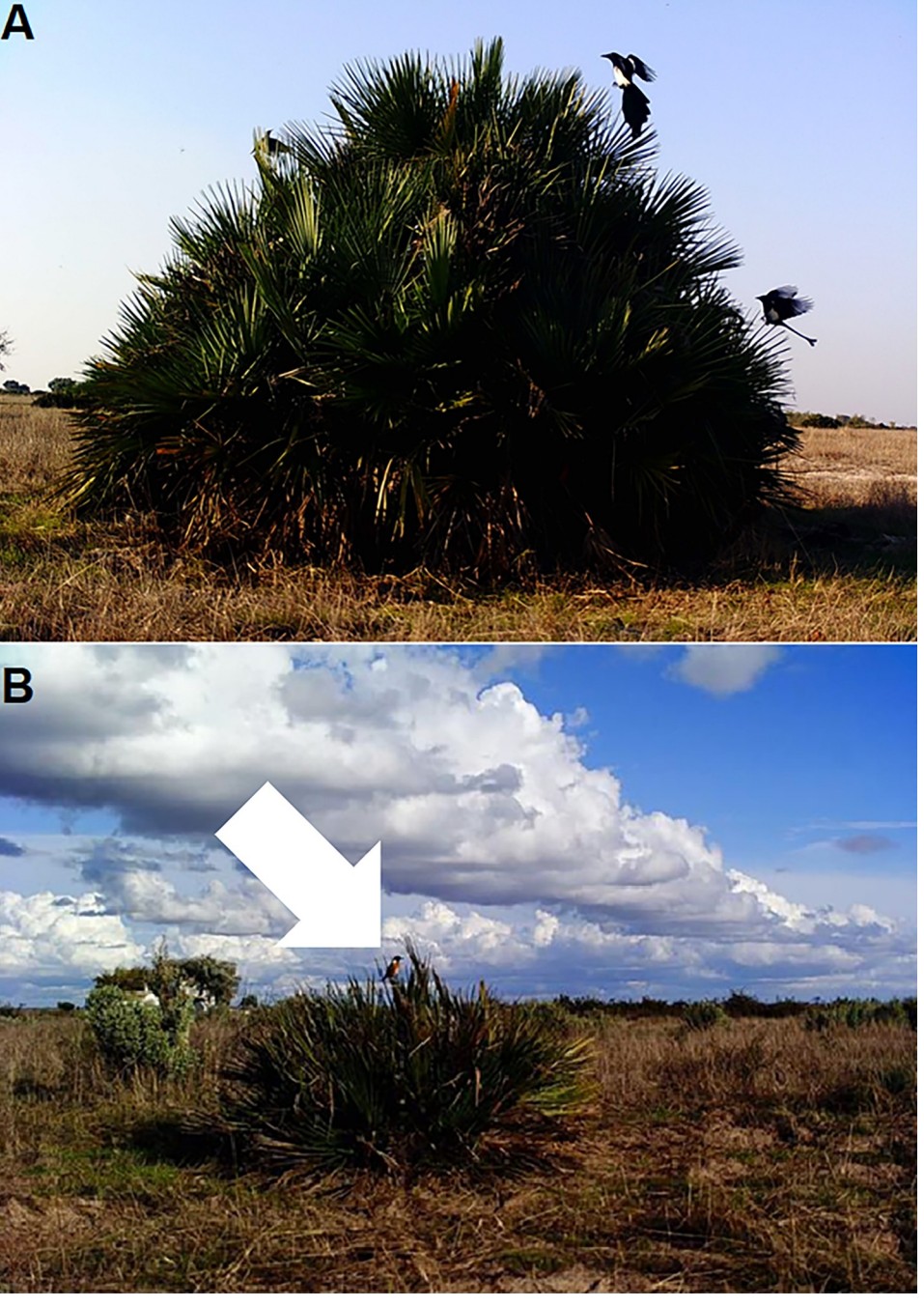

**Fig 2. Birds using *Chamaerops humilis* as perches in Doñana National Park.** (A) Eurasian magpie (*Pica pica*). (B) Common stonechat (*Saxicola rubicola*). (Photo credit: Pedro J. Garrote).

senator, *Phoenicurus ochruros, Phylloscopus collybita, Pica pica, Saxicola rubicola, Sturnus unicolor, Sylvia atricapilla* and *Sylvia melanocephala* (Fig 2, Table 1).

Every target *C. humilis* was checked for bird feces and regurgitations by carefully checking the palm foliage surface. We also evaluated whether seeds arrive at open spaces without palms. To this aim, we checked for the presence of bird feces and regurgitations in an open area associated with each focal *C. humilis*. These areas were of similar surface and mostly located in

**Table 1. Summary of collected data.**

| | EARLY-SUCCESSIONAL PLOT | | LATE-SUCCESSIONAL PLOT | |
|---|---|---|---|---|
| | **2019** | **2020** | **2019** | **2020** |
| *Chamaerops humilis* (individuals) | 109 | | 180 | |
| Number of feces | 705 | 206 | 1560 | 527 |
| Number of seeds | 28 | 16 | 340 | 145 |
| *C. humilis* with feces (%) | 42.2 | 37.6 | 43.9 | 27.8 |
| Feces with seeds (%) | 3.3 | 17.8 | 7.5 | 7.8 |
| *C. humilis* with seeds (%) | 8.3 | 11.9 | 13.9 | 7.8 |
| *Asparagus aphyllus* (seeds) | 1 | 7 | 16 | 3 |
| *Daphne gnidium* (seeds) | 7 | 5 | 1 | 2 |
| *Olea europaea* var. *sylvestris* (seeds) | 0 | 0 | 26 | 8 |
| *Phillyrea angustifolia* (seeds) | 0 | 0 | 4 | 0 |
| *Pistacia lentiscus* (seeds) | 20 | 4 | 2 | 0 |
| *Rubus ulmifolius* (seeds) | 0 | 0 | 291 | 132 |
| **Perching birds** | | | | |
| *Cisticola juncidis* | 1 | 1 | 0 | 2 |
| *Lanius senator* | 0 | 0 | 2 | 0 |
| *Phoenicurus ochruros* | 0 | 0 | 3 | 1 |
| *Phylloscopus collybita* | 0 | 0 | 1 | 0 |
| *Pica pica* | 0 | 0 | 2 | 0 |
| *Saxicola rubicola* | 2 | 1 | 7 | 4 |
| *Sturnus unicolor* | 0 | 0 | 2 | 0 |
| *Sylvia atricapilla* | 1 | 1 | 1 | 2 |
| *Sylvia melanocephala* | 3 | 3 | 14 | 4 |
| **Beneficiary species** | | | | |
| *Asparagus aphyllus* | 35 | | 96 | |
| *Cistus sakviifolius* | 16 | | 0 | |
| *Daphne gnidium* | 1 | | 0 | |
| *Olea europaea* var. *sylvestris* | 6 | | 20 | |
| *Pistacia lentiscus* | 3 | | 0 | |
| *Pyrus bourgaeana* | 1 | | 0 | |
| *Quercus suber* | 1 | | 5 | |
| *Rubus ulmifolius* | 3 | | 7 | |
| *Stauracanthus genistoides* | 31 | | 17 | |

Number of samples (feces, seeds) collected in each study plot, the number of seeds found for each plant species, as well as the number and species of bird recorded perching on *C. humilis*.

bare soils with scarce vegetation (mainly grasses) and 3–4 m from the associated *C. humilis*. Every found excrement and regurgitated seed was collected and stored inside of an individual Eppendorf tube. In the lab, feces were dried to make their handling easier. Once dried, every sample was checked for seeds by crumbling the excrement and looking for hard matter, either of vegetable (seeds, bark) or animal (elytra and other arthropod parts) origin. Found seeds were identified at the species level using images of a reference collection (Fig 3), considering its shape, size and the surface design. All collected seeds (N = 529) were positively identified.

To assess the potential selection by birds of particular traits of *C. humilis* individuals, we measured the following palm variables: gender (male, female, undetermined), size (height,

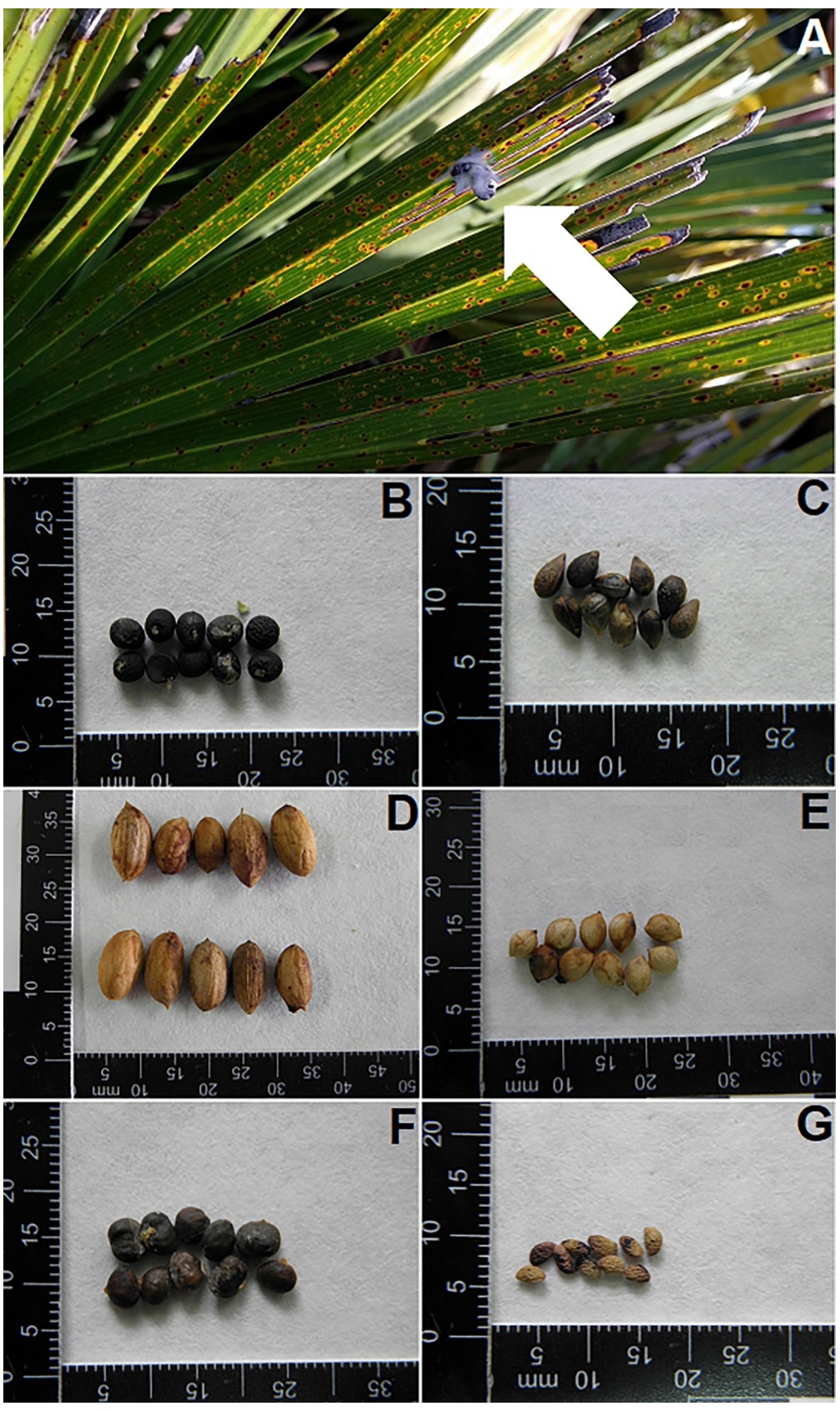

**Fig 3. Examples of collected feces and identified dispersed seeds.** (A) Bird excrement on a leaf of *Chamaerops humilis* in the early-successional plot. Seeds of (B) *Asparagus aphyllus*, (C) *Daphne gnidium*, (D) *Olea europaea* var. *sylvestris*, (E) *Phillyrea angustifolia*, (F) *Pistacia lentiscus*, and (G) *Rubus ulmifolius*. (Photo credit: Pedro J. Garrote).

area and volume), presence and richness of beneficiary woody species associated with them and beneficiary height (< 25 cm, 25–100 cm, > 100 cm). The beneficiary species were woody plants that grow under dwarf palms and whose branches appear from the upper surface of these. These species often benefit from *C. humilis* due to microclimatic improvement and protection against herbivory [42].

## Spatial analyses

**Aggregation of *C. humilis* used as perches within the overall pattern *of C. humilis*.** Every *C. humilis* was characterized with its spatial location and a mark (presence/absence of bird feces in this case). To find out if *C. humilis* used as perches by birds (i.e. receiving bird feces) show a spatial pattern (aggregated, isolated) or they instead are randomly distributed we used qualitatively marked point patterns techniques and the random labelling null models to evaluate lack of any spatial structure. The random labelling null model consists in giving the mark "with feces" or "without feces" over all *C. humilis* (having or not feces) randomly, conducting a total of 199 simulations [38].

In order to quantify the spatial pattern of *C. humilis* used as perches within the overall pattern of *C. humilis* we used mark connection functions [43–45] as summary statistics [38]. Whether *C. humilis* contained feces they were assigned with a mark of value 1 and a mark of value 0 if they had no feces. The mark connection function $p_{ij}(r)$ gives the probability that two *C. humilis*, separated by a distance *r*, the first is type *i* (i.e. with feces) and the second is type *j* (without feces).

Finally, to test if *C. humilis* with feces are located in areas of overall high density of neighbouring *C. humilis* (with feces or without them) we used the test statistic $g_{i,i+j}(r) - g_{j,i+j}(r)$ [44], which consist in comparing the density of *C. humilis* (*i* + *j*) around *C. humilis* with feces (type *i*) with the density of *C. humilis* (*i* + *j*) around *C. humilis* without feces (type *j*). Under random labelling, the expected value for this statistic is zero: in case that feces tend to be delivered in palm aggregations we expected $g_{i,i+j}(r) > g_{j,i+j}(r)$ and, if not, if there is an isolation of feces, the we expected $g_{i,i+j}(r) < g_{j,i+j}(r)$.

To test the fit of data with the point process and departures from the null model, we conducted 199 simulations (for every analysis) of the fitted point processes and estimated envelopes with an approximate error rate of $\alpha = 0.05$ [46] which are the fifth lowest and highest values of the summary statistics of the simulated point process. Observed values above the top or below the bottom simulation envelopes indicate higher or lower than expected spatial aggregation, respectively. Observed values within the simulation envelopes indicate a level of spatial aggregation compatible with the stochasticity of the point process model. For all point pattern analyses, we used the grid-based software *Programita* [47].

**Aggregation of *C. humilis* receiving dispersed seeds within the overall pattern of *C. humili*.** To assess if *C. humilis* containing (feces with) dispersed seeds were randomly distributed within the pattern of all *C. humilis* (with and without seeds) or they follow a particular spatial pattern (aggregation, dispersion), we also used qualitatively marked point pattern techniques and the random labelling null model to represent lack of any spatial structure. *The same process and analysis were conducted as in the previous case but, in this occasion, the used mark was the presence (1) or absence (0) of seed in each individual C. humilis. In parallel, these same analyses were carried out but excluding the Rubus' seeds since their great number could*

*distort the spatial pattern for the presence of seeds on C. humilis (see* S2 File *and* S2 Fig*). Since results were rather consistent whether we considered or not Rubus's seeds, here we provide the results when all seed species are included.*

**Association of the number of dispersed seeds on *C. humilis*.** To find out whether the number of dispersed seeds within bird feces deposited on *C. humilis* has a spatial association among different *C. humilis* we used quantitative marked point patterns techniques and the random labelling null model to represent lack of any spatial structure.

To quantify potential spatial associations in the number of seeds on different *C. humilis* we used the univariate mark correlation function as summary statistic [38, 43, 47, 48], ergo we wanted to find out if closer *C. humilis* had a higher number of seeds than isolated ones. In addition to its location, each *C. humilis* was characterized by a quantitative mark: the number of dispersed seeds of any species.

Mark correlation functions are based on pairs of study units (in this case *C. humilis* individuals) and the idea of estimating the mean value of a test function $t(m_i, m_j)$ of the two marks $m_i$ and $m_j$, taken over all the ordered pairs $i$-$j$ of *C. humilis* which have interpoint distances of $r \pm h$, being $h$ the parameter bandwidth, which must be wide enough to produce a sufficient number of pairs in each distance class r but small enough to reveal significant biological detail [43, 49]. From all possible test functions $t(m_i, m_j)$ [43], we selected the following three:

(*i*) The r-mark correlation function $k_{m1.}(r)$ which is based on the test function: $t(m_i, m_j) = m_i$. $k_{m1.}(r) > 1$ indicates that the number of seeds on *C. humilis* that have neighbouring *C. humilis* at distance r is larger than the average of seeds, showing a positive effect of *C. humilis* aggregation on seed number. By contrast, if $k_{m1.}(r) < 1$, it would indicate a negative effect of aggregation and a positive effect of isolation of *C. humilis* on seed number [49]. Finally, if $k_{m1.}(r) \sim 1$, indicates that the number of seeds is not affected by the distance to *C. humilis* individuals.

(*ii*) Another mark correlation function was calculated to characterize the spatial covariance in number of seeds of two *C. humilis* separated by distance $r$. This test function is known as Schlather's: $t(r, m_i, m_j) = [m_i—\mu(r)][(m_j—\mu(r)]$ which results in a Moran's I like summary statistic $I_{mm}(r)$, a spatial variant of the Pearson correlation coefficient between the variables $m_i$ and $m_j$ defined by the ordered $i$-$j$ pairs of *C. humilis* separated by distance $r \pm h$.

(*iii*) Lastly, we used a function that relates the number of seeds on a *C. humilis* directly to the density of neighbouring *C. humilis*, called "density correlation function" [49]: $C_{m,K}(r)$. This function estimates the Pearson correlation coefficient between the number of seeds $m_i$ and the number of neighbours within distance $r$, based on the following equation: $t(r, m_i, K_i) = [m_i = \mu][(\lambda K_i(r)—\lambda K_i(r)]$. Where $m_i$ is the number of seeds of the focal *C. humilis* $i$, $\lambda$ is the overall density of *C. humilis* in the study area, $\mu$ is the mean number of seeds of the population, $\lambda K_i(r)$ is the number of neighbours around the focal *C. humilis* $i$ within distance $r$ for all *C. humilis*.

To test whether the data show spatial correlation, the observed values of the three functions mentioned above were contrasted to the data obtained from simulations of a null model without spatial correlation in the marks. The null model consisted in shuffling randomly the marks (number of seeds) over all *C. humilis*, conducting a total of 199 simulations. These simulations were also useful to estimate the simulation envelopes for the three functions, being the fifth highest and lowest values of the summary statistics among all simulations of the null mode [38, 49].

## Effect of palm traits

To analyse how our response variables (presence of bird feces, presence of dispersed seeds and number of dispersed seeds) were influenced by the size (height, area and volume) and gender (male, female) of *C. humilis* individuals, as well as by the presence (and height: < 25 cm, 25–100 cm, >100 cm) of beneficiary species, three sets of generalized linear models (GLMs) were fitted per study plot.

Model 1 assessed the effect of palm traits on the likelihood of presence of bird feces, Model 2 evaluated the probability of finding dispersed seeds depending on such characteristics and Model 3 appraised the number of seeds based on those characteristics. Both Model 1 and Model 2 have a binary response variable (presence = 1, absence = 0), and thus were fitted with a binomial distribution and a logit link. Finally, Model 3, was fitted with a Poisson distribution with logit link.

A set of competing models were fitted and the best candidate model was selected based on Akaike's Information Criterion (AIC), allowing to compare different hypothesis and know their supportive level on the data by weighting them (AICc; [50]). All analyses were performed in R 4.0.3 statistical software [51]. Additionally, these same models were carried out but using the interaction between *C. humilis* Height (m) and Area (m$^2$) instead of the variable Volume (m$^3$). Results were rather consistent with those from the previous analysis (see S2 Table).

## Results

### Overall patterns

We studied total of 109 *C. humilis* in the early-successional plot, from which were collected 705 bird feces and 28 seeds in 2019 and 206 bird feces and 16 seeds in 2020 (Fig 4). In the late-successional plot, we monitored 180 *C. humilis*, 1560 feces and 340 seeds in 2019, and 527 feces and 145 seeds in 2020. Nevertheless, not all dispersed seeds were contained within the feces since a small fraction was regurgitated by the birds. Six shrub species were identified among the collected seeds: *A. aphyllus*, *D. gnidium*, *O. europaea* var. *sylvestris*, *P. angustifolia*, *P. lentiscus* and *R. ulmifolius*. Finally, a large fraction of *C. humilis* acted as perch for birds and contained their feces: 42.2% (2019) and 37.6% (2020) in the early-successional plot and 43.9% (2019) and 27.8% (2020) in the late-successional plot (Table 1).

Besides, most *C. humilis* had beneficiary woody plants growing within them. In the late-successional plot and the early-successional plot 104 out of 180 individuals (57.78%) and 71 out to 109 (65.14%) had beneficiary plants, respectively. A total of nine different beneficiary species were found, all of them present in the early-successional plot but just five were recorded in the late-successional plot (Table 1).

### Spatial patterns

**Aggregation of *C. humilis* used as perches within the overall pattern of *C. humilis*.** Analysis of how *C. humilis* used as perches by birds (i.e. receiving bird feces) were distributed within the overall pattern of *C. humilis* showed the following results. In the early-successional plot, the probability that two *C. humilis*, separated by a distance *r*, both used as perches (statistic $p_{11}(r)$), fall within the expectation of random labelling (Fig 5A). The probability that two separated *C. humilis*, with just one comprising feces (statistic $p_{12}(r)$), was included within the simulation envelops generated by random labelling (Fig 5C). Finally, the observed density of neighbouring *C. humilis* (regardless of having feces or not; statistic $g_{i,i+j}(r)-g_{j,i+j}(r)$) around *C. humilis* with feces was within the simulation envelops generated by a random labelling mode (Fig 5E). Thus, in the early-successional plot, the observed probabilities described by the three

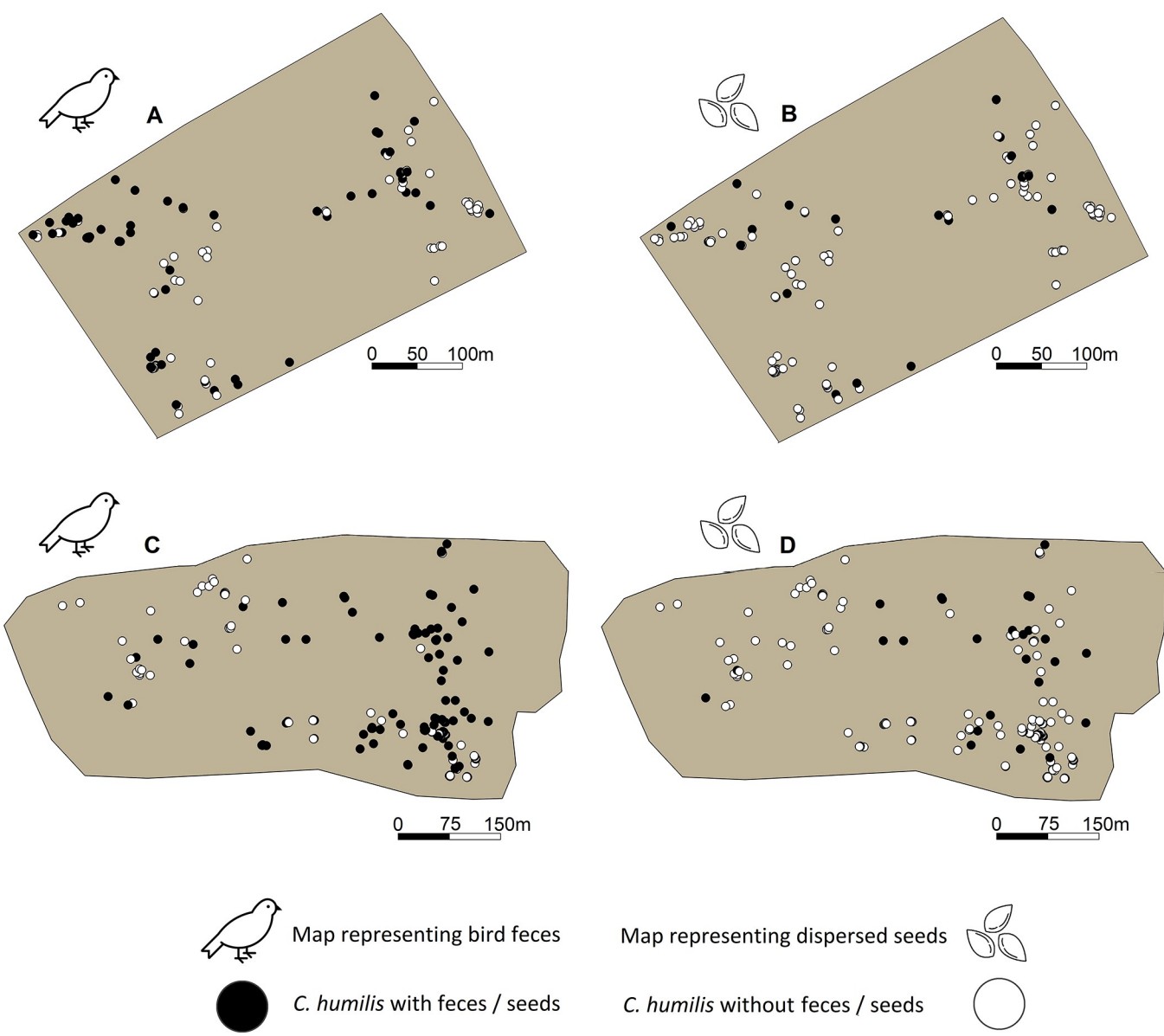

**Fig 4. Distribution of *C. humilis* in the early-successional plot (upper) and the late-successional plot (lower) plots.** (A) *C. humilis* in the early-successional plot with (black solid points) and without (white circles) bird feces or regurgitations. (B) *C. humilis* in the early-successional plot with (black solid points) and without (white circles) dispersed seeds. (C) *C. humilis* in the late-successional plot with (black solid points) and without (white circles) bird feces. (D) *C. humilis* in the late-successional plot with (black solid points) and without (white circles) dispersed seeds.

statistics were within the envelops generated by random labelling models, which indicates they are compatible with the stochasticity of the process model.

On the contrary, in the late-successional plot, the probability that two *C. humilis* separated for distances 1-6m, both having feces ($p_{11}(r)$ statistic), was lower than the expectation under random labelling. The bivariate $p_{12}(r)$ showed that *C. humilis* with and without feces were further away than expected under random labelling up to scales of 6m (Fig 5D). Finally, the density of neighbouring *C. humilis* (having or not feces) around *C. humilis* with feces was significantly lower than expected by random labelling for distances up to scales of 7m (Fig 5F).

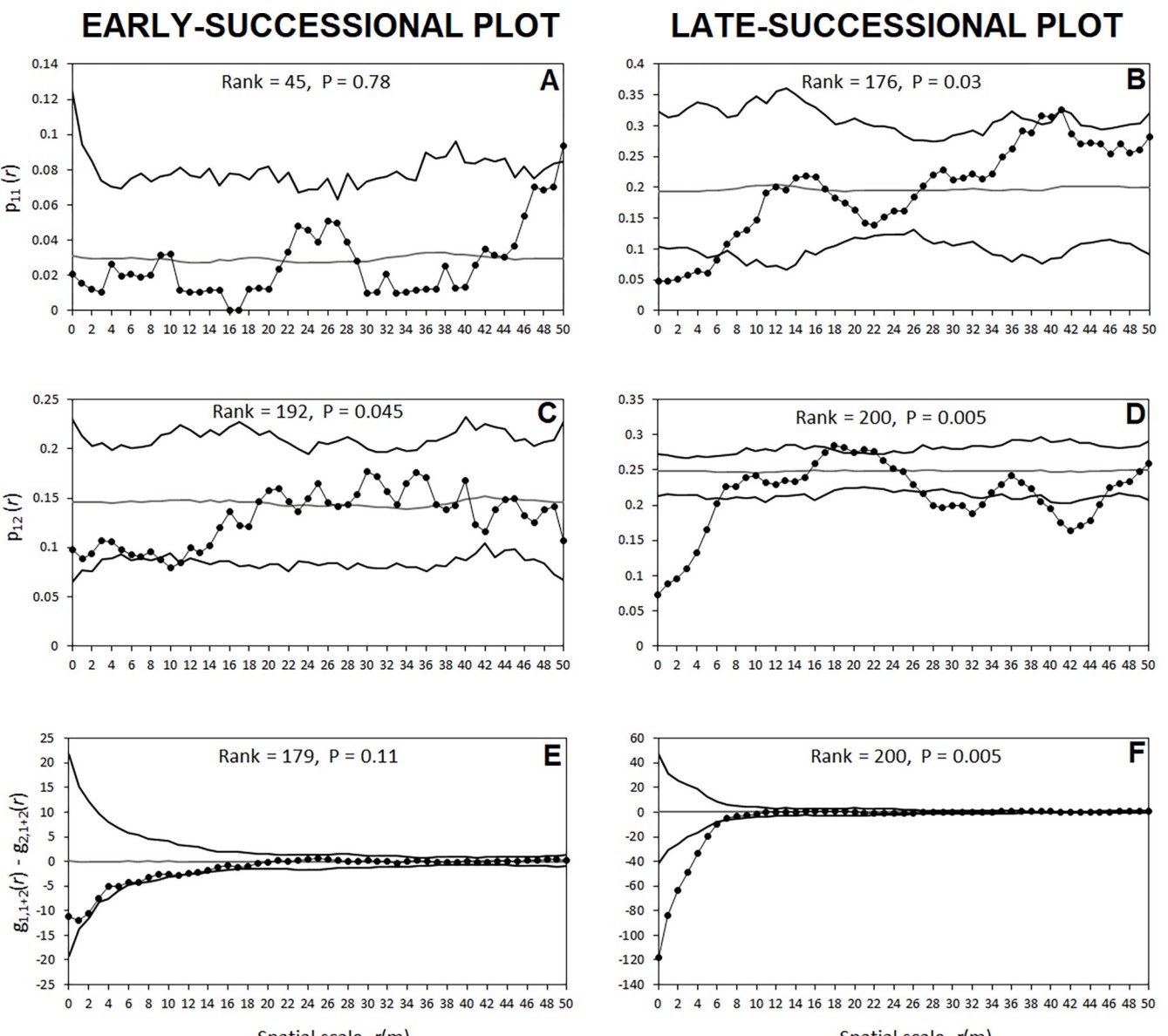

**Fig 5. Analysis of *C. humilis* with feces in the two study plots (the early and late-successional plots) areas using mark connection functions as summary statistics.** (A, B) The mark connection function $p_{11}(r)$ gives the conditional probability that, from two *C. humilis* that are separated by distance *r*, both are type 1 (i.e., with feces). (C, D) The mark connection function $p_{12}(r)$ gives the conditional probability that, from two *C. humilis* that are separated by distance *r*, the first is type 1 (i.e., with feces) and the second is type 2 (i.e., without feces). (E, F) The test statistic $g_{1,1+2}(r)—g_{2,1+2}(r)$ compares the density of *C. humilis* (i.e., 1 +2) around *C. humilis* with feces (i.e., type 1) with the density of *C. humilis* (i.e., 1+ 2) around *C. humilis* without feces (i.e., type 2). The expected mark connection function statistics (gray line) and the corresponding simulation envelopes (black lines), being the fifth lowest and highest values of the functions created by 199 simulations under random labelling, are also shown.

Overall, these results indicate a tendency by frugivorous birds in the late-successional plot towards using and defecating in isolated *C. humilis* individuals.

**Distribution of *C. humilis* receiving dispersed seeds within the overall pattern of *C. humilis*.**   In the early-successional plot, the probability $p_{11}(r)$, where two *C. humilis* separated by distance *r*, both having seeds, was contained within the envelopes of random labelling for all spatial scales (Fig 6A). The bivariate $p_{12}(r)$, the probability for two neighbouring *C. humilis*, one with seeds and other without them, was also contained between random labelling's

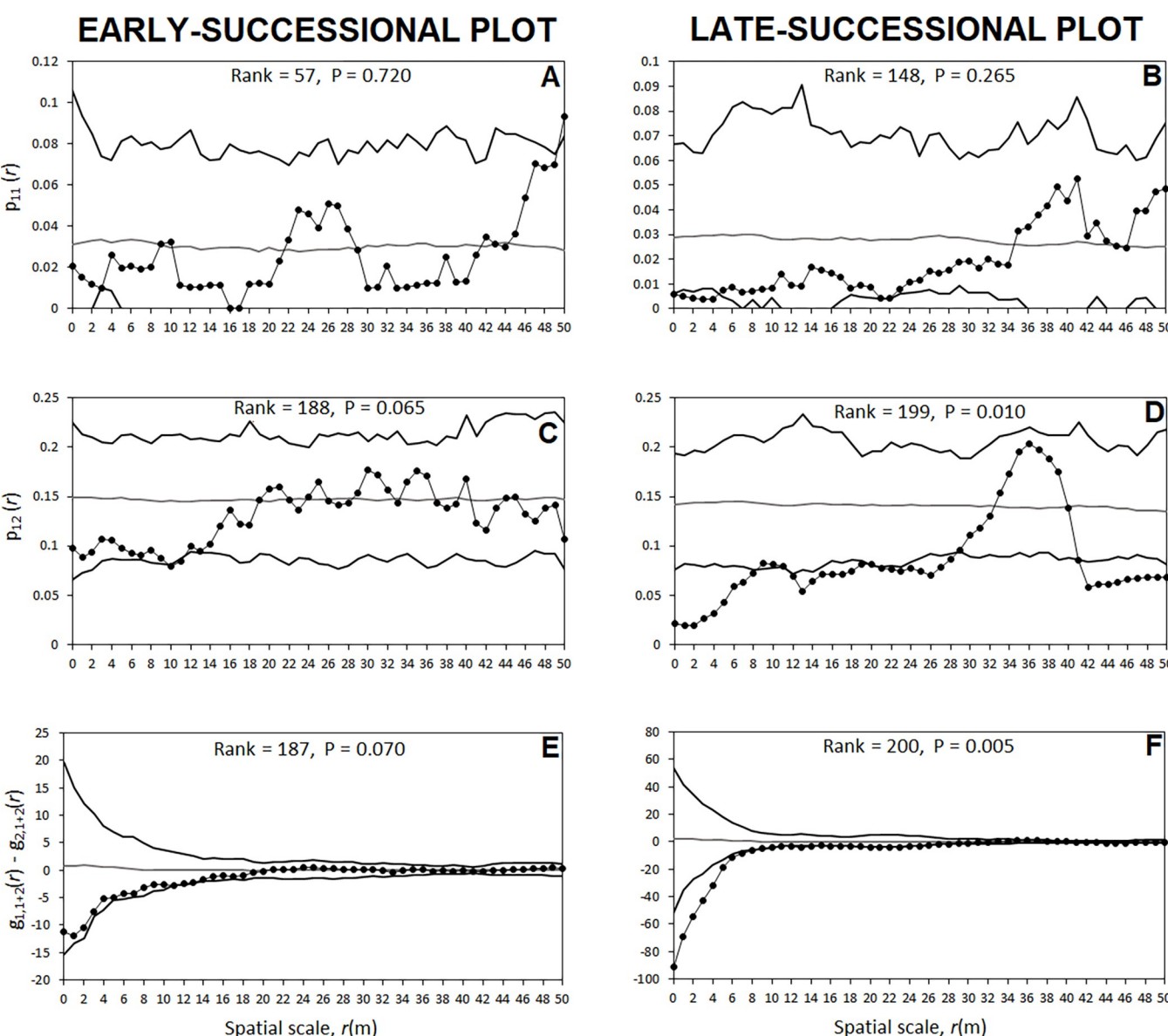

**Fig 6. Analysis of feces with seeds in the two study plots (the early and late-successional plots) using mark connection functions as summary statistics.**
(A, B) The mark connection function $p_{11}(r)$ gives the conditional probability that, from two *C. humilis* that are separated by distance *r*, both are type 1 (i.e., with seeds). (C, D) The mark connection function $p_{12}(r)$ gives the conditional probability that, from two *C. humilis* that are separated by distance *r*, the first is type 1 (i.e., with seeds) and the second is type 2 (i.e., without seeds). (E, F) The test statistic $g_{1,1+2}(r)$—$g_{2,1+2}(r)$ compares the density of *C. humilis* (i.e., 1+2) around *C. humilis* with seeds (i.e., type 1) with the density of *C. humilis* (i.e., 1+ 2) around *C. humilis* without seeds (i.e., type 2). The expected mark connection function statistics (gray line) and the corresponding simulation envelopes (black lines), being the fifth lowest and highest values of the functions created by 199 simulations under random labelling, are also shown.

envelopes (Fig 6C). Finally, the test statistic $g_{1,1+2}(r)$—$g_{2,1+2}(r)$ shows that the density of neighbouring *C. humilis* around *C. humilis* receiving seeds did not differ from the mean *C. humilis* density. (Fig 6E).

Results in the late-successional plot substantially differed from results from the early-successional plot. The probability of two *C. humilis* separated by distance *r*, having both seeds ($p_{11}(r)$ statistic), was similar to expected of random labelling for all spatial scales (Fig 6B). However, the probability for two neighbouring *C. humilis*, just one of them with seeds ($p_2(r)$

statistic), was significantly lower (*P* = 0.01) that the obtained by using random labelling for distances up to 6 m, between 12 and 29 m and higher than 41 m (Fig 6D). Neighbourhood density of *C. humilis* around *C. humilis* receiving seeds was significantly lower (*P* = 0.005) than the mean *C. humilis* density at scales of 1 to 6 m (Fig 6F).

In short, the analysis of how *C. humilis* with seeds were distributed within the overall pattern of *C. humilis* showed that *C. humilis* with seeds were randomly distributed in Matagordas (Fig 6A, 6C and 6E) while, in the late-successional plot, *C. humilis* with seeds were spatially isolated at small spatial scale (Fig 6D and 6F).

**Distribution of the number of dispersed seeds across *C. humilis*.** The observed r-mark correlation function for the number of seeds was included within the simulation envelop for all scales, indicating no relationship between distance among individual *C. humilis* and the number of dispersed seeds neither in the early-successional plot (*P* = 0.95, Fig 7A) nor in the late-successional plot (*P* = 0.31, Fig 7B). Similarly, the Schlather's correlation function did not show a correlation between number of seeds and distance among individuals at any spatial scales, neither in the early-successional plot (*P* = 0.22, Fig 7C) nor in the late-successional plot (*P* = 0.22, Fig 7D). Finally, the density correlation function did not indicate correlation between the number of seeds and the density of *C. humilis* at any scale neither in the early-successional plot (*P* = 0.35, Fig 7E) nor in the late-successional plot (*P* = 0.35, Fig 7F).

## Effect of palm traits

We found that *C. humilis* in the early-successional plot were on average (8.80 m$^2$) smaller than in the late-successional plot (15.26 m$^2$). Also, the richness of beneficiary species beneath *C. humilis* was slightly lower in the early-successional plot than in the late-successional plot, with 0.89 and 1.07 beneficiary species per individual, respectively.

For the early-successional plot, the best model (AIC = 133.97) showed that presence of bird feces on *C. humilis* were positively related to their height and area, but negatively to volume, although none of these variables were found significant (Table 2. Model 1). The best model for the late-successional plot (AIC = 196.63) indicated that presence of bird feces on *C. humilis* was higher in male than females and positively influenced by palm area, but negatively to the richness of beneficiary species contained within them. Nevertheless, only the palm area was found significant (*P* < 0.001).

Presence of dispersed seeds in the early-successional plot depended on *C. humilis* gender and height (Table 2. Model 2), according to the best model obtained (AIC = 96.343). Nonetheless, only the height was significant (*P* < 0.05) with larger *C. humilis* individuals having a higher number of dispersed seeds. In the late-successional plot, the best model obtained (AIC = 137.98) showed that height and richness of beneficiary species had a positive effect on the number of dispersed seeds, although just the last one was significant in this case (*P* < 0.01) (Table 3. Model 2).

According to the best model in the early-successional plot (AIC = 191.41), male *C. humilis* had a higher number of dispersed seeds than females (*P* < 0.05) and height had a positive and significant (*P* < 0.001) influence on the number of seeds found on *C. humilis*. Palm volume had a negative effect on the number of seeds (*P* < 0.05) (Table 2. Model 3). Finally, in the late-successional plot, just one model was fitted (AIC = 1273.09). This model predicted: (*i*) a negative relationship between height and area of *C. humilis* and the number of dispersed seeds (*P* < 0.0001); (*ii*) a higher number of dispersed seeds in male *C. humilis* than in females (*P* < 0.0001); (*iii*) a positive relationship between *C. humilis* volume and the number of dispersed seeds (*P* < 0.0001); (*iv*) a positive relationship between the richness of beneficiary species in *C. humilis* and the number of dispersed seeds (*P* < 0.0001); and (*v*) a higher number of

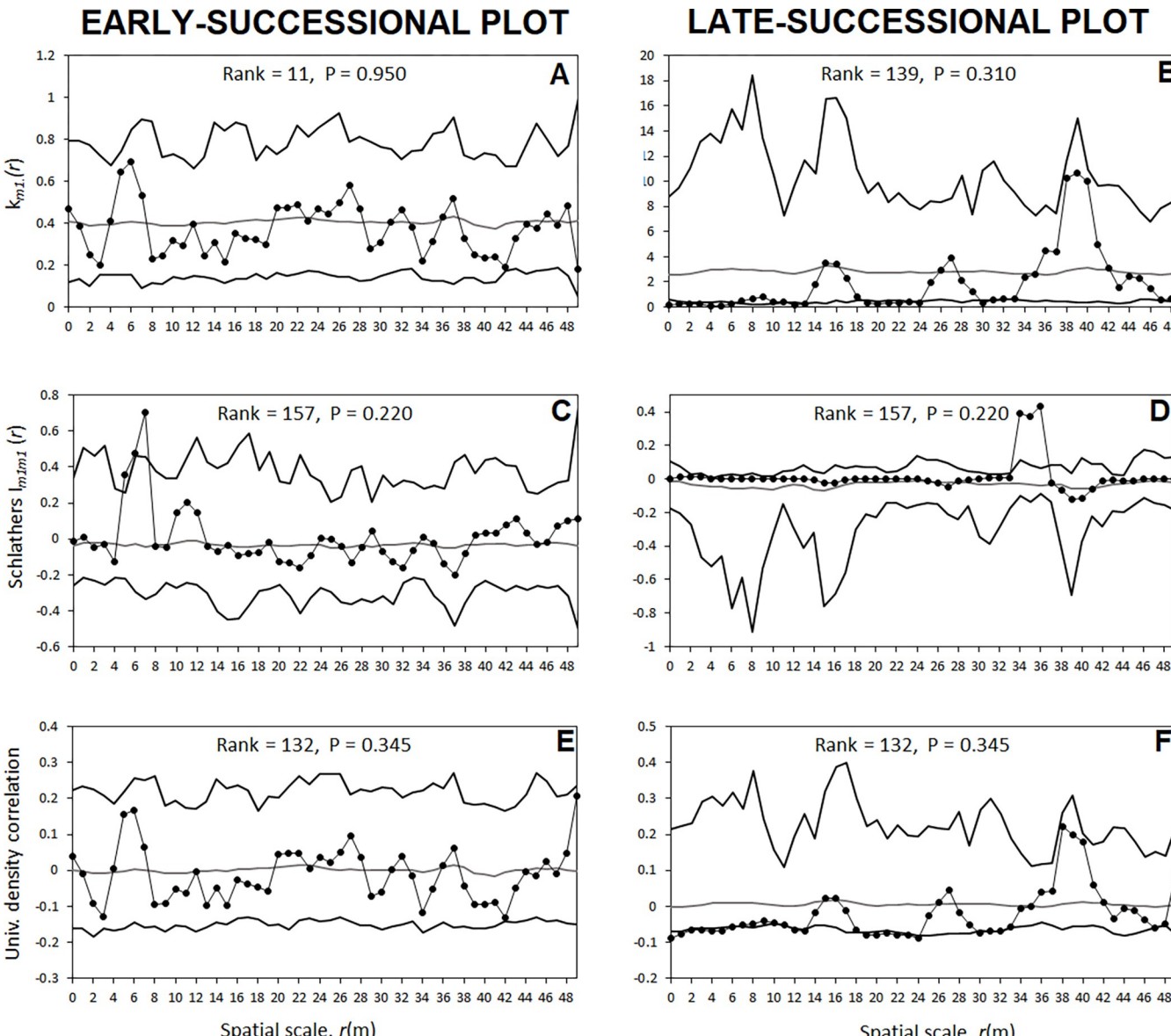

**Fig 7. Mark correlation functions to evaluate a potential spatial structure in the number of dispersed seeds found on *C. humilis* in the early and late-successional study plots.** (A, B) The r-mark correlation function describes the mean number of seeds ($m_i$) on a *C. humilis* at distance r of another *C. humilis*. (C, D) Schlather's correlation function quantifies the correlation between the number of seeds on two different *C. humilis* separated by distance r. (E, F) Density correlation function assesses the correlation between the number of seeds and the number of neighbours located at a distance r. The expected mark connection function statistics (gray line) and the corresponding simulation envelopes (black lines), being the fifth lowest and highest values of the functions created by 199 simulations under random labelling, are also shown.

dispersed seeds in *C. humilis* holding tall (> 100 cm in height) beneficiary woody individuals as compared to both small (< 25 cm) and intermediate (25–100 cm) beneficiary individuals ($P < 0.05$; Table 3, Model 3).

## Discussion

The role of pioneer plants acting as perches for seed dispersers is crucial for the restoration of many human-disturbed habitats by increasing seed arrival and eventually seedling

**Table 2. GLM carried out to analyze the palm traits in the early-successional plot.**

| Explanatory variable | Competing models | | β | SE | AICc | ΔAIC | Weighted AIC |
|---|---|---|---|---|---|---|---|
| *Model 1* | | | | | | | |
| Presence of Bird Feces | Height (m) + Area (m²) + Volume (m³) | | | | 133.97 | 0 | 0.62 |
| | | Intercept* | -4.586 | 2.244 | | | |
| | | Height (m) | 5.930 | 3.513 | | | |
| | | Area (m²) | 0.411 | 0.219 | | | |
| | | Volume (m³) | -0.450 | 0.282 | | | |
| | Gender + Height (m) + Area (m²) + Volume (m³) | | | | 135.67 | 1.7 | 0.27 |
| | Gender + Height (m) + Area (m²) + Volume (m³) + Richness | | | | 137.54 | 3.57 | 0.10 |
| | Gender + Height (m) + Area (m²) + Volume (m³) + Richness + Beneficiary height (cm) | | | | 143.11 | 9.14 | 0.01 |
| Deviance = 90.343 | | | | | | | |
| *Model 2* | | | | | | | |
| Presence of Dispersed Seeds | Gender + Height (m) | | | | 96.343 | 0 | 0.05 |
| | | Intercept* | -4.969 | 1.441 | | | |
| | | Gender (Male) | 0.915 | 0.557 | | | |
| | | Height (m)* | 4.294 | 1.837 | | | |
| | Gender + Height (m) + Volume (m³) | | | | 98.21 | 1.87 | 0.02 |
| | Gender + Height (m) + Area (m²) + Volume (m³) | | | | 99.03 | 2.69 | 0.01 |
| | Gender + Height (m) + Area (m²) + Volume (m³) + Richness | | | | 100.73 | 4.39 | < 0.01 |
| | Gender + Height (m) + Area (m²) + Volume (m³) + Richness + Beneficiary height (cm) | | | | 107.56 | 11.2 | < 0.01 |
| Deviance = 126.52 | | | | | | | |
| *Model 3* | | | | | | | |
| Number of Dispersed Seeds | Gender + Height (m) + Volume (m³) | | | | 191.41 | 0 | 0.11 |
| | | Intercept* | -5.988 | 1.153 | | | |
| | | Gender (Male)* | 0.822 | 0.354 | | | |
| | | Height (m)* | 7.737 | 2.033 | | | |
| | | Volume (m³)* | 0.134 | 0.065 | | | |
| | Gender + Height (m) + Area (m²) + Volume (m³) | | | | 192.77 | 1.36 | 0.06 |
| | Gender + Height (m) + Area (m²) + Volume (m³) + Richness | | | | 194.47 | 3.06 | 0.02 |
| | Gender + Height (m) + Area (m²) + Volume (m³) + Richness + Beneficiary height (cm) | | | | 196.18 | 4.77 | 0.01 |
| Deviance = 133.75 | | | | | | | |

Summary of fitted parameters and models employed to analyse the presence of bird feces (model 1), the presence of dispersed seeds (model 2) and the number of dispersed seeds (model 3) on *C. humilis* in the early-successional study plot. Competitive models are ranked from the lowest AICc value (best model) to the highest one, being significant variables indicated with asterisks (*).

establishment [52–55]. Using a spatially explicit approach, we detected a random spatial structure in both the number of visits (bird feces) and the dispersed seeds found on *C. humilis* in one study plot (the early-successional) and an aggregated pattern in the other one (the late-successional). Frugivorous birds tended to disperse seeds into a small number of *C. humilis* individuals, usually male and large-sized ones, that acted as "hotspots" of seed arrival. Our study also revealed that the species of dispersed seeds partially matched the woody beneficiary

**Table 3. GLM carried out to analyze the palm traits the late-successional plot.**

| Explanatory variable | Competing models | | β | SE | AICc | ΔAIC | Weighted AIC |
|---|---|---|---|---|---|---|---|
| **Model 1** | | | | | | | |
| Presence of Bird Feces | Gender + Area (m²) + Richness | | | | 196.63 | 0 | 0.54 |
| | | Intercept* | -0.776 | 0.304 | | | |
| | | Gender (Male) | 0.5583 | 0.352 | | | |
| | | Area (m²)* | 0.089 | 0.024 | | | |
| | | Richness | -0.485 | 0.266 | | | |
| | Gender + Area (m²) + Volume (m³) + Richness | | | | 197.88 | 1.25 | 0.29 |
| | Gender + Height (m) + Area (m²) + Volume (m³) + Richness | | | | 199.49 | 2.86 | 0.13 |
| | Gender + Height (m) + Area (m²) + Volume (m³) + Richness + Beneficiary height (cm) | | | | 201.76 | 5.13 | 0.04 |
| Deviance = 188.63 | | | | | | | |
| **Model 2** | | | | | | | |
| Presence of Dispersed Seeds | Height (m) + Richness | | | | 137.98 | 0 | 0.62 |
| | | Intercept* | -3.383 | 0.549 | | | |
| | | Height (m) | 0.480 | 0.273 | | | |
| | | Richness* | 1.971 | 0.731 | | | |
| | Height (m) + Volume (m³) + Richness | | | | 139.86 | 1.88 | 0.24 |
| | Height (m) + Area (m²) + Volume (m³) + Richness | | | | 141.64 | 3.66 | 0.10 |
| | Gender + Height (m) + Area (m²) + Volume (m³) + Richness | | | | 143.5 | 5.52 | 0.04 |
| | Gender + Height (m) + Area (m²) + Volume (m³) + Richness + Beneficiary height (cm) | | | | 147.44 | 9.46 | < 0.01 |
| Deviance = 131.98 | | | | | | | |
| **Model 3** | | | | | | | |
| Number of Dispersed Seeds | Gender + Height (m) + Area (m²) + Volume (m³) + Richness + Beneficiary height (cm) | | | | 1273.09 | 0 | 1 |
| | | Intercept* | -2.823 | 1.078 | | | |
| | | Gender (Male)* | 1.802 | 0.127 | | | |
| | | Height (m)* | -1.946 | 0.458 | | | |
| | | Area (m²)* | -0.114 | 0.014 | | | |
| | | Volume (m³)* | 0.134 | 0.014 | | | |
| | | Richness* | 0.957 | 0.093 | | | |
| | | Beneficiary height (25-100cm) | 0.7794 | 1.034 | | | |
| | | Beneficiary height (>100cm)* | 2.59102 | 1.022 | | | |
| Deviance = 1155.7 | | | | | | | |

Summary of fitted parameters and models employed to analyse the presence of bird feces (model 1), the presence of dispersed seeds (model 2) and the number of dispersed seeds (model 3) on *C. humilis* in the late-successional study plot. Competitive models are ranked from the lowest AICc value (best model) to the highest one, being significant variables indicated with asterisks (*).

species growing inside *C. humilis*, reinforcing the idea that this palm species plays a key role in the restoration of disturbed habitats.

## Widespread perching effect by *C. humilis*

Considering that our sampling period did not take place throughout the year and that an unknown fraction of the bird feces presents in target *C. humilis* individuals surely went

unnoticed, the data obtained could be considered as a hint of how widespread the "perching" effect in both study plots. Therefore, the high presence of (frugivorous) bird feces and dispersed seeds in both study plots is noteworthy.

Besides, it is apparent that some seeds can emerge seedlings beneath *C. humilis*, since all seeds collected belonged to beneficiary woody species found within *C. humilis*. Nevertheless, we did not find seeds of the beneficiary species *C. salviifolius*, *P. bourgaeana*, *Q. suber* and *S. genistoides*, which also are frequently associated with *C. humilis*. Most of these plant species are seldom dispersed by birds. Other animals are more likely to disperse them, such as insects for *C. salviifolius* and *S. genistoides* seeds [56–58] or mammals, such as ungulates (red deer, wild boars, domestic cattle) and carnivores (Eurasias badger, red fox) for *P. bourgaeana* seeds [59]. Therefore, the presence of these woody species associated with *C. humilis* reveals that it does not attract just bird-dispersed plants but also mammal and insect-dispersed plants, accentuating its role in habitat restoration.

## Spatial patterns of the perching effect

Our study has revealed different spatial patterns in both study plots at the three target organizational levels studied: (*i*) *C. humilis* receiving bird feces, (*ii*) *C. humilis* receiving (feces with) dispersed seeds, and (*iii*) number of dispersed seeds arriving to *C. humilis*.

No spatial pattern was found in the early-successional plot, where the distribution of *C. humilis* did not seem to play a decisive role in attracting frugivorous birds. By contrast, in the late-successional plot, where isolated palms were more frequent than in the early-successional plot, there was strong evidence that birds preferred isolated *C. humilis*, at least for those separated from conspecifics by a distance up to 7 m. Aggregated perches could be found attractive for seed dispersers and, therefore, more visited by them [60], since they would act as shelters against predators [61]. This premise has been used for designing ecological restoration and conservation actions in mixed ecosystems, where trees or other woody plants, acting as perches and nurses, are planted following aggregated spatial patterns [62–65]. However, by concentrating frugivorous activity to aggregated planted trees, seed dispersal could be limited [16, 55]. Therefore, our results from the late-successional plot supporting the role of isolated perches diverge from the most widespread tendency in ecological restoration of aggregating trees and shrubs during revegetation efforts [65].

Spatial patterns of perching effect differed between study plots. On the one hand, these differences could be attributed to the changing distance that frugivorous birds must fly between the main seed source area (Mediterranean scrubland) and our study plots where target *C. humilis* individuals are located. This distance is longer in the early-successional plot (up to 350 m) than in the late-successional plot (150–200 m). Besides, it seems reasonable to expect that birds select *C. humilis* located on the edge of the plot, since these individuals are closer to the Mediterranean scrubland and thus allow birds to move between habitats with a minimal consumption of energy. On the other hand, both plots are open habitats and have been intensively disturbed by humans in the past, but with some decades apart [16, 41]. Thus, study plots are in different stages of ecological succession and the vegetation surrounding them is different too. For instance, in the early-successional plot there are more scattered trees (mostly *Q. suber* and *O. europaea* var. *sylvestris*) than in the late-successional plot. The reduced number of feces and seeds and the lack of spatial patterns found in the early-successional plot could relate to the bird's intensive usage of such scattered trees. A recent study has shown that in the later-successional site the "nursing" effect of *C. humilis* may be diluted among different shrub species [4], since the landscape is dominated by shrub-encroached communities. The "perching" effect of *C. humilis*, however, seems to follow the opposite trend, being enhanced in the most shrub-encroached landscape (i.e. Reserva) probably because of the few available trees for perching.

Interestingly, we found that most of dispersed seeds were concentrated in a small number of *C. humilis* individuals that acted as "hotspots" of seed arrival. For example, in the late-successional plot there was a single large *C. humilis* individual receiving 47.4% of the total seeds found in that plot (N = 485). These results are supported by previous evidence that a small number of individual trees can play an important role in habitat restoration by altering the movements of seed dispersers and thus increasing influx of animal-dispersed seeds [16, 55]. Features (other than spatial patterns) that make particular *C. humilis* individuals hotspots of seed arrival are discussed below.

## Palm trait's influence on perching effect

This investigation identified some features that made particular *C. humilis* attractive "perching" sites for frugivorous birds and, therefore, hotspots for seed arrival. However, this pattern was found only in the late-successional plot, and not in the early-successional plot. In the late-successional plot, larger *C. humilis* individuals received more seeds as compared to small ones. The greater surface of large individuals could attract more birds since it implies a greater number of entry points to *C. humilis* interior, where the bird could be safe from predators and adverse weatherly conditions (e.g. high temperatures, wind).

The use of *C. humilis* as "perching" site by birds does not necessarily implies seed arrival since not all recorded birds are typical seed dispersers [66]. However, we sampled during peak of the seed dispersal season, time when most recorded birds change their diet into a more fruit-based one. For instance, it has been reported that *P. ochruros*, *S. atricapilla* and *S. malenocephala* are able to quickly change their diet when fleshy fruits become available, while other species such as *S. rubicola* show a more gradual change to a frugivorous diet [67, 68]. Also, *S. unicolor*, *P. ochruros*, *P. pica*, and *S. rubicola* has been reported as fruit-eater, although in lesser quantities [69–73]. Therefore, although not all recorded bird species contribute equally to seed dispersal, fleshy fruits represent a variable fraction of their diets during the seed dispersal season.

Secondly, the preference of taller *C. humilis* by frugivorous birds could relate to their own requirements as well as to characteristics of the surrounding landscape [74]. Taller *C. humilis* could suppose safer places since they would provide greater visual fields to birds to spot potential predators. Also, taller perches are likely better sites to take flight towards other habitats (e.g. the Mediterranean scrubland) adjacent to the studied site, which may be more suitable habitats for them [64, 75, 76].

The detected positive effect of richness of beneficiary species within *C. humilis* in the late-successional plot could be related to some of them producing fleshy fruits (e.g. *Rubus ulmifolius*, *Asparagus aphyllus*), acting thus as an additional attractant for frugivorous birds [77–79]. Richness of beneficiaries was also found positively related to the number of seeds found on *C. humilis*, which is likely a consequence of the increased number of frugivorous bird visits. Additionally, beyond the fact of bearing fleshy fruits, the own beneficiary species could be used as a perch and thus attract birds. Therefore, a similar reasoning explained earlier about how *C. humilis* height influences seed dispersal, could be applied for the revealed positive effect of height of beneficiary plants. Finally, it seemed that male *C. humilis* were more likely to receive dispersed seeds than females. We speculate that male *C. humilis* could be selected by birds over females because the former have a lower associated predation risk. Females *C. humilis* produce fleshy fruits that are too large for birds, but they are often eaten by medium-sized carnivores such as the Eurasian badger and the red fox [33]. Although these mammals are mainly nocturnal, their presence likely prevents birds from using female *C. humilis* in fruit as shelter at night-time, and thus may dissuade them at daytime too. Therefore, the use of *C. humilis* in ecological

restoration should follow a balance between male and female plants. Male plants would attract preferably birds that would disperse small-seeded woody species present in the surrounding area, while female plants would attract medium-sized carnivores and ungulates that would mostly disperse large-seeded species such as *P. bourgaena* and *C. humilis*.

In summary, taller and isolated male *C. humilis* with a higher diversity of beneficiary species seemed to be the most suitable individuals for attracting frugivorous birds and, therefore, received higher number of dispersed seeds. However, this pattern was not entirely confirmed in the early-successional plot, likely because of a much smaller sample sizes (i.e. lesser numbers of both *C. humilis* used as perches and of dispersed seeds).

## Conclusions and implications for ecological restoration

In the last 30 years, restoration ecology has undergone considerable development thanks to the emerging new tools and techniques that facilitate the retrieval of biological properties of disturbed habitats [80]. Our study plots are a good representation of Mediterranean old fields, which have been targets for habitat restoration [5, 16]. Previous studies have demonstrated the value of *C. humilis* in these programs since it facilitates recruitment of numerous woody species dispersed by frugivores [4, 36]. Here, we have demonstrated that *C. humilis* also exerts a marked "perching" effect, crucial in restoration and ecological succession [27, 52, 81, 82]. The combination of artificial and natural perches with nurse plants could increase the seed arrival and recruitment leading to nucleation processes around them [4, 19, 20]. In this sense, to our knowledge, this study represents the first spatially explicit assessment of the relationship between natural perches and seed arrival via frugivorous birds in Mediterranean human-disturbed landscapes. Our results could be used by managers to decide, for example, whether particular areas are adequate to promote seed arrival using natural or artificial perches.

Our finding in the late-successional plot indicating that dispersed seeds are concentrated in a few *C. humilis* individuals, which tend to be isolated from the rest, breaks surprisingly with the general trend of revegetating under an aggregation design (e.g. [64, 65, 83]). However, our results are in agreement with Fedriani and collaborators [16] whose simulated *P. bourgaeana* seed dispersal into human-altered habitats within the Doñana National Park and found that planting isolated trees was the most efficient strategy to enhance seed arrival. Though further research is needed to evaluate the pervasiveness of our findings, the strategy of using aggregated perches for increasing seed arrival and habitat restoration likely should not be advocated under all ecological circumstances.

To conclude, this investigation supports previous studies about the importance of the pioneer *C. humilis* for the restoration of human-disturbed Mediterranean landscapes. We prove for the first time its key role as natural perch, an ecological function that, though it is shared with some woody plants, is especially relevant in early stages of ecological succession when few pioneer species occur. Birds tended to disperse seeds into a small number of *C. humilis* individuals (usually isolated male and large-sized ones) that acted as "hotspots" of seed arrival. Thanks to the double role of *C. humilis*, which attracts frugivorous birds that deliver considerable numbers of seeds of late-successional woody species and provides improved conditions for seedling emergence and survival, it is likely to become a successful tool for ecological restoration in many habitats of the western Mediterranean basin.

## Supporting information

**S1 File. Cluster analysis.** Technical details concerning the cluster analysis.
(DOCX)

**S2 File. Spatial analysis excluding *Rubus'* seeds.**
(DOCX)

**S1 Table. Summary of the results of the *C. humilis* distribution fitted with the Thomas cluster process.**
(DOCX)

**S2 Table. GLM carried out to analyze the palm traits in the early-successional plot.** A General Linear Model (GLM) was carried out with the interaction between *C. humilis* Height (m) and Area (m$^2$) as a new variable instead of the *C. humilis* Volume (m$^3$). The results found are pretty similar to the ones obtained by using the variable Volume (m$^3$) in both study plots. Thus, we decided to maintain the initial model.
(DOCX)

**S3 Table. GLM carried out to analyze the palm traits in the late-successional plot.** A General Linear Model (GLM) was carried out with the interaction between *C. humilis* Height (m) and Area (m2) as a new variable instead of the *C. humilis* Volume (m3). The results found are pretty similar to the ones obtained by using the variable Volume (m3) in both study plots. Thus, we decided to maintain the initial model.
(DOCX)

**S1 Fig.** Univariate cluster analysis for *Chamaerops humilis* in the early (A, C, E, G) and late-successional study plots (B, D, F, H). (A, B) Pair correlation function $g$(r). (C, D) *L*-function *L* (r). (E, F) Spherical contact distribution $H_S$(r). (G, H) Nearest neighbor distribution function $D1$(r). The expected mark connection function statistics (gray line) and the corresponding simulation envelopes (black lines), being the fifth lowest and highest values of the functions created by 199 simulations of the null model, are also shown.
(DOCX)

**S2 Fig. Analysis of feces with seeds (excluding *Rubus*) in the late-successional study plot using mark connection functions as summary statistics.** (A) The mark connection function $p_{11}(r)$ gives the conditional probability that, from two *C. humilis* that are separated by distance *r*, both are type 1 (i.e., with seeds). (B) The mark connection function $p_{12}(r)$ gives the conditional probability that, from two *C. humilis* that are separated by distance *r*, the first is type 1 (i.e., with seeds) and the second is type 2 (i.e., without seeds). (C) The test statistic $g_{1,1+2}(r)$—$g_{2,1+2}(r)$ compares the density of *C. humilis* (i.e., 1 + 2) around *C. humilis* with seeds (i.e., type 1) with the density of *C. humilis* (i.e., 1 + 2) around *C. humilis* without seeds (i.e., type 2). (D, E, F) Mark correlation functions to evaluate a potential spatial structure in the number of dispersed seeds (excluding *Rubus*). (D) The r-mark correlation function describes the mean number of seeds (mi) on a *C. humilis* at distance r of another *C. humilis*. (E) Schlather's correlation function quantifies the correlation between the number of seeds on two different *C. humilis* separated by distance r. (F) Density correlation function assesses the correlation between the number of seeds and the number of neighbours located at a distance r. The expected mark connection function statistics (gray line) and the corresponding simulation envelopes (black lines), being the fifth lowest and highest values of the functions created by 199 simulations under random labelling, are also shown.
(DOCX)

## Author Contributions

**Conceptualization:** Pedro J. Garrote, Jose M. Fedriani.

Formal analysis: Víctor González-García.

Investigation: Víctor González-García, Pedro J. Garrote, Jose M. Fedriani.

Methodology: Víctor González-García, Pedro J. Garrote, Jose M. Fedriani.

Resources: Pedro J. Garrote, Jose M. Fedriani.

Supervision: Pedro J. Garrote, Jose M. Fedriani.

Writing – original draft: Víctor González-García.

Writing – review & editing: Víctor González-García, Pedro J. Garrote, Jose M. Fedriani.

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
