## [Decision Letter · Decision Letter 0]

25 Feb 2022

PONE-D-22-01669Unmasking the perching effect of the pioneer Mediterranean palm *Chamaerops humilis* L.PLOS ONE

Dear Dr. González-García,

Thank you for submitting your manuscript to PLOS ONE. After careful consideration, we feel that it has merit but does not fully meet PLOS ONE’s publication criteria as it currently stands. Therefore, we invite you to submit a revised version of the manuscript that addresses the points raised during the review process.

This paper examines an interesting restoration question about the role of perch plants for seed dispersal in disturbed environments.  It addresses an important conservation issue but the paper needs substantive improvements to be suitable for publication, including editing for language issues.  The questions are clearly delineated, although, as the reviewers suggest there is some detail missing in the description of the design and initial predictions.  The results are interesting and well organized but need more clarification, as Reviewer #2 details. 

The discussion, though, needs substantial revision to increase clarity and improve the interpretation of your results, especially in terms of language.  It contains many awkwardly worded sentences that are challenging to understand (e.g., lines 459-462; 465-467; 493-497; 508-511; 550-552; 565-566; 594-596) and/or use poor grammar (e.g., lines 268-269; 502-503; 536-537; 559-561).   Some of these issues could be minimized if you use more active rather than passive voice.

In particular, it will be important to address the concerns of the both of the reviewers who highlight a number of areas that need further exploration.  Reviewer #1 has some concerns about the foraging and movement ecology of the dispersers and the potential impact on the seed dispersal patterns as well as the large differences in the types of seeds potentially dispersed.  Please also note that Reviewer #1 has some specific questions that should be addressed to improve clarity, particularly in the study design and motivations.  Reviewer #2 also provides some valuable recommendations for each section and identifies a number of places where more explanation or revision is needed.  Some additional issues are identified below:

Line 63: “template” should be “temperate”

Line 99: The use of the term “beneficiary” requires some type of description/definition or a reference source.  What characteristics define this group and is it context-dependent?

Line 232: Why 199 simulations?  It seems like a rather arbitrary number.  Is this based on some initial modeling or preliminary results?

Line 255: “that” should be “than”

Both reviewers provide detailed suggestions in how to improve the manuscript and reframe the paper to make it better highlight the strengths.  With substantial improvements this paper could make a welcome contribution to the restoration literature.

We look forward to receiving your revised manuscript.

Kind regards,

Karen Root, Ph.D.

Academic Editor

PLOS ONE

Journal Requirements:

5. We note that Figures 1 and 4 in your submission contain map images which may be copyrighted. All PLOS content is published under the Creative Commons Attribution License (CC BY 4.0), which means that the manuscript, images, and Supporting Information files will be freely available online, and any third party is permitted to access, download, copy, distribute, and use these materials in any way, even commercially, with proper attribution. For these reasons, we cannot publish previously copyrighted maps or satellite images created using proprietary data, such as Google software (Google Maps, Street View, and Earth). For more information, see our copyright guidelines: http://journals.plos.org/plosone/s/licenses-and-copyright.

 a. You may seek permission from the original copyright holder of Figures 1 and 4 to publish the content specifically under the CC BY 4.0 license. 

Reviewers' comments:

Reviewer's Responses to Questions

**Comments to the Author**

1. Is the manuscript technically sound, and do the data support the conclusions?

Reviewer #1: Partly

Reviewer #2: Partly

2. Has the statistical analysis been performed appropriately and rigorously? 

Reviewer #1: Yes

Reviewer #2: Yes

3. Have the authors made all data underlying the findings in their manuscript fully available?

Reviewer #1: Yes

Reviewer #2: Yes

4. Is the manuscript presented in an intelligible fashion and written in standard English?

Reviewer #1: No

Reviewer #2: Yes

5. Review Comments to the Author

Reviewer #1: Although there is a lot of consideration on the use of artificial perches in restoration, the role of natural perches deserves more investigation, especially in a poorly studied biome under this perspective, as is the case with Mediterranean Europe. In this regard, the authors provide an important dataset based on field observations and fine spatial analyses, which represents a great contribution to the restoration of that particular habitat.

Despite their efforts to explain the different patterns found between the two study sites (Mastasgordas and Reserva), I think this attempt was a bit confused, perhaps because there may be underlying causes not fully investigated by the authors. I would put this question in the perspective of the field data and not in the analysis itself. One underlying problem may be the unbalanced frequency of birds and seeds in the two areas. From the nine bird species recorded, six are reputed insectivores and just three can be considered omnivores that can take significant amounts of fruit in their diets (https://esajournals.onlinelibrary.wiley.com/doi/abs/10.1890/13-1917.1). These three omnivore species performed more than 50% of the visits recorded in the two areas and probably deposited more seeds per visit beneath the palms than the other ones. Special attention should be paid to the foraging and movement patterns of these three omnivores, since they may account for the general pattern of the seed rain and, perhaps, provide sound explanations for these patterns.

Another point addresses the plant composition and fruit types between the two areas. Rubus ulmifolius is the only recorded fruit consumed that is a functional berry, containing dozens of seeds. The other plant species are single-seeded drupes. This species was not related as common or abundant in Matasgordas, whose seed rain was nil, but its presence was highlighted in Reserva, whose seed rain under C. humilis accounted for 85-90% of the total seed rain. Even a few feces with dozen of R. ulmifolius seeds would certainly bias the results and the comparison between areas when the response variable “number of seeds” is considered. Would the spatial patterns be the same between the areas if R. ulmifolius is excluded from the analysis? If so, the conclusions presented by the authors should be revisited and adjusted to the bias introduced by this species.

I see these points do not invalidate the importance of C. humilis as perches and facilitative species in that habitat, but at least would provide more explanatory power for the distinct spatial patterns found between the two study sites. Besides that, I would recommend the authors to improve the English text, as some sentences are somewhat difficult to understand.

Bellow, I add some other specific points to the authors.

Page 4, lines 94-96: To question whether there are differences in the role of C. humilis as perches between the two areas presupposes some previous knowledge of the distributional pattern of this species between the areas. Although the authors provide a detailed description of the study areas in the proper session (pages 6 and 7) it should be enlightening to give a hint here on what triggered the idea that the plant role could be different between areas. Certainly, these two areas were not chosen at random.

Page 6, line 142: Delete the period after bourgaeana.

Page 7, lines 177-178: What was the total sampling effort for bird perching for the separate areas?

Page 9, lines 189-192: What do you mean by “palm interior”? Does it include the foliage surface in the interior and also the soil under the foliage? Was the open area of a similar surface associated with the palms bare soil? What were the criteria used to distinguish between bird-and-mammal dispersed seeds in these areas?

Page 14, lines 312-313: Since the number of feces and number of seeds are two interrelated response variables, what is the weight of the regurgitated seeds in the analysis? How much regurgitated seeds contributed to the total number of seeds sampled?

Page 16, line 375: Consider this writing: “Otherwise, in Reserva, the probability …”

Page 2, line 501: Insert “of” after “chances”.

Reviewer #2: Review MS:

Unmasking the perching effect of the pioneer Mediterranean palm Chamaerops humilis L.

I have now carefully read the manuscript. I found the authors approach very interesting using SPPA to unravel the perching effect. Using this spatial analysis they evaluate several spatial patterns of “perching” effect and whether the seed arrival via frugivorous birds is related to the spatial distribution of the C. humilis in two disturbed scenarios. I think the manuscript could be interesting will be interesting for readers interested in restoration or ecological succession drivers.

I think the manuscript needs to improve the readability mostly in the methods and the results section (see comments below). I have concerns with the general work. On one hand, I think that it needs more information about the state of the selected plots. For instance, the main objective is to evaluate the perching effect based on the plants distribution, however, the authors did not show the spatial distribution of the individual plants, plants density, size of the plot, etc. that in my opinion has to be in the article. On the other hand, your main conclusions are based on some patterns found in one plot (male plants, isolation). This is my major concern because when you present the analysis of the other plot the main patterns did not applied to other ecological context. It seems that the results only applied under certain circumstances. I think this is a very common problem when studying the effects of perching which makes it very difficult to generalize the drivers of this process.

Abstract

L39- “dispersed seeds We detected” missed a point

Introduction

L97: Do exist some individuals of C. humilis that act as “hotspots” of dispersed seeds? Change for: Do any C. humilis individuals act as "hot spots" for seed dispersal?

Methods

L134-144. With this explanation it is unclear where did you work.

Reading some works made in the area it seems that Matasgordas has a higher plant density than Reserva. For instance, In the article by Fedriani and Delibes (2011) when the description of the Matasgordas site is made, it seems to have a higher density of individuals, same situation in the article by Jácome et al. 2016 where the spatial patterns of this species are studied where they also report higher number of individuals in the area. Thus, it would be interesting to specify that you do it “in a low density area in order to be comparable with Reserva.”

L163- Related with my previous comment. Although Reserva was protected earlier authors have to consider that there still traditional uses in the area, thus plants in the Reserva are submitted to a high herbivorous pressure due to the presence of horses, cattle and high density of cervids. This pressure has reflected in the lower density of individuals explained above.

Suggested papers:

-Fedriani, J. M., & Delibes, M. (2011). Dangerous liaisons disperse the Mediterranean dwarf palm: Fleshy- pulp defensive role against seed predators. Ecology, 92, 304–315.

-Jácome-Flores, M. E., Delibes, M., Wiegand, T. & Fedriani, J. M. Spatial patterns of an endemic Mediterranean palm recolonizing old fields. Ecol. Evol. 6, 8556–8568 (2016).

L170-Every target C. humilis individual (109 and 180 in Matasgordas and Reserva, respectively)

SPPA requires the delimitation of an area in which all target individuals of the study species are georeferenced. I suppose this was the case, however it is not clear in the methodology. I suggest to put more information such as density and/or no. of individuals, area sampled, etc. for each selected plot. Also this information will clarify more the differences that you found between plots.

Discussion

L463-L478- Reduce the length of this paragraph, there is too much mention of dispersal mechanisms of other species. This section should only focus on the ability of C. humilis to attract different dispersers including mammals.

L484-485 I suggest to the authors to do SPPA to the individual plants to unravel the spatial patterns of the “Matasgordas and Reserva C. humilis populations”, this analysis could be useful to explain some of the patterns. For instance maybe the patterns of the plants in Matasgordas tend to be more random explaining your findings without any over explanation. On the other hand maybe Reserva population its characterized by more isolated plants that had enough height to function as a perch for birds

L528-529: …while the surface of Chamaerops humilis was found significant in Reserva

I suggest to change surface to cover area

L526-534. Sometimes is difficult and confusing to follow the explanations or results because sometimes appear first Matasgordas and then Reserva. I suggest to the authors to be consistent with the appearance order through all document

L550 Change: What is more to Furthermore or Additionally

L552- Unexpectedly, it was found that larger C. humilis (i.e. those with larger volume) were less suitable for seed arrival, at least in Reserva.

-Change “it was found” to “we found”

-I suggest to the authors not use volume but instead the interaction between cover area (surface) and height. This interaction could reveal some perching preferences regarding this two variables. The variety of shapes in this palm make it difficult to standardize the volume as a “common variable” between all individuals.

L554-We speculate that male C. humilis could be selected over females because they lack fruits. C. humilis fruits are too big for birds, but they are often eaten by medium-sized carnivores such as the European badger and the red fox [33]. The presence of these predators could dissuade frugivorous birds from visit female C. humilis in fruit, and thus perching more often male palms.

I suggest to put some information about the abundance of “potential predators” in each study area to reinforced this idea. Another explanation to this pattern (in my opinion more likely) is the presence of rodents and rabbits beneath the female plants. Rodents and rabbits are also highly attracted to C. humilis fruits, in this foraging process they could predate the seeds dropped in the perching process, limiting the number of seeds. In this context, the same situation could be happening in aggregated C. humilis individuals that are preferred by rodents due to high vegetation cover.

L564- It seems that your results could be useful for restoration planification where the C. humilis is a very useful species. However the inconsistency of the results between areas and the unclear relation between the traits masked by the context dependency seems to be an issue to the applicability of the study. For instance the efficiency of the perching effect seems to be related to male plants, if we use this results as it comes we would be deprecated female plants that would be useful to attract super-efficient fleshy fruit dispersers such as badgers or foxes.

6. PLOS authors have the option to publish the peer review history of their article (what does this mean?). If published, this will include your full peer review and any attached files.

Reviewer #1: No

Reviewer #2: No

---

## [Author Response · Author response to Decision Letter 0]

31 May 2022

Dear Dr. Karen Root, 

We have now prepared a substantially revised version of our manuscript "Unmasking the perching effect of the pioneer Mediterranean palm Chamaerops humilis L." (PONE-D-22-01669) submitted to Plos One. We would like to thank you and the reviewers for your thoughtful comments, and we are happy with the careful and constructive criticisms and helpful suggestions. In response to the comments, we have introduced the following three main changes: 

 1. As suggested by the editor and both reviewers, the manuscript has been carefully revised several times to improve the English grammar and flow of logic. 

 2. Following suggestion by reviewer #1, the role of birds as frugivorous and seed dispersers have been described in further detail. Also, some spatial analyses have been repeated excluding data from Rubus ulmifolius and the result are now provided in a new appendix.

 3. As suggested by reviewer #2, we have conducted cluster analyses to estimate the distribution of C. humilis in both study plots. Also, we fitted a new GLM to estimate the effect of the interaction between palm height and palm area on seed arrival to C. humilis. Results of these new analyses are provided in the corresponding new appendices. 

Please, find below an itemized list detailing how we have responded to each of the items in the reports. The comments are in black font, our response follows in blue font, and changed phrases in response to the comment are provided in blue font and between quotation marks and.

Thank you again for your kind assistance and the time devoted to our manuscript.

Sincerely,

V.

 

1. Changes made to address editor comments:

The discussion, though, needs substantial revision to increase clarity and improve the interpretation of your results, especially in terms of language. It contains many awkwardly worded sentences that are challenging to understand (e.g., lines 459-462; 465-467; 493-497; 508-511; 550-552; 565-566; 594-596) and/or use poor grammar (e.g., lines 268-269; 502-503; 536-537; 559-561). Some of these issues could be minimized if you use more active rather than passive voice.

Thank you for your suggestions. To address them, we made the following changes:

Lines 495-499 (previously 459-462): “Considering that our sampling period did not take place throughout the year and that an unknown fraction of the bird feces presents in target C. humilis individuals surely went unnoticed, the data obtained could be considered as a hint of how widespread the "perching” effect in both study plots. Therefore, the high presence of (frugivorous) bird feces and dispersed seeds in both study plots is remarkable.”.

Lines 501-504 (previously 465-467): “Nevertheless, we did not find seeds of the beneficiary species C. salviifolius, P. bourgaeana, Q. suber and S. genistoides, which are usually associated to C. humilis. Most of these plant species are seldom dispersed by birds”.

Lines 516-519 (previously 493-497): “However, by concentrating frugivorous activity to aggregated planted trees, seed dispersal could be limited [16, 55]. Therefore, our results supporting the role of isolated perches diverge from the most widespread tendency in ecological restoration of aggregating trees and shrubs during revegetation efforts [65]”.

Lines 523-526 (previously 508-511): “However, by concentrating frugivorous activity to aggregated planted trees, seed dispersal could be limited [16, 55]. Therefore, our results supporting the role of isolated perches diverge from the most widespread tendency in ecological restoration of aggregating trees and shrubs during revegetation efforts [65]”.

Lines 583-587 (previously 550-552): “Additionally, beyond the fact of bearing fleshy fruits, the own beneficiary species could be used as a perch and thus attract birds. Therefore, a similar reasoning explained earlier about how C. humilis height influences seed dispersal, could be applied for the revealed positive effect of height of beneficiary plants”.

Line 605-607: (previously 565-566): “In the last 30 years, restoration ecology has undergone considerable development thanks to the emerging new tools and techniques that facilitate the retrieval of biological properties of disturbed habitats [80]”. 

Lines 635-638 (previously 594-596): “We prove for the first time its key role as natural perch, an ecological function that, though it is shared with some fleshy-fruited woody plants, is especially relevant in early stages of ecological succession when few pioneer species occur”.

Lines 288-289 (previously 268-269): “Finally, if km1. (r) ~ 1, indicates that the number of seeds is not affected by the distance to C. humilis individuals”.

Lines 531-533 (previously 502-503): “Besides, it seems reasonable to expect that birds select C. humilis located on the edge of the plot, since these individuals are closer to the Mediterranean scrubland and thus allow birds to move between habitats with a minimal consumption of energy”.

Previous lines 536-537: deleted.

Lines 598-600 (previously 559-561): “In summary, taller (and isolated) male C. humilis with a higher diversity of beneficiary species seemed to be the most suitable for attracting frugivorous birds and, therefore, received higher number of dispersed seeds”.

In particular, it will be important to address the concerns of the both of the reviewers who highlight a number of areas that need further exploration. Reviewer #1 has some concerns about the foraging and movement ecology of the dispersers and the potential impact on the seed dispersal patterns as well as the large differences in the types of seeds potentially dispersed. Please also note that Reviewer #1 has some specific questions that should be addressed to improve clarity, particularly in the study design and motivations. Reviewer #2 also provides some valuable recommendations for each section and identifies a number of places where more explanation or revision is needed. Some additional issues are identified below:

Regarding Reviewer #1’s concerns: we have improved our manuscript according to the comments received. We answered the questions related to the ecology of dispersers and did some new analyses without considering the Rubus’ seeds. We found that the results obtained when we excluded these seeds barely differed from the results obtained when those seed were included. 

In respect of Reviewer #2’s comments, we have considerably improved the clarity and readability of the whole manuscript. We also added some information that Reviewer #2 considered important to clarify some aspects (C. humilis distribution, replicability, area of study, origin and identification of feces, etc.). We also run a new GLM including the interaction between C. humilis height (m) and area (m2) instead of the variable volume (m3), as suggested. The GLM obtained for both study plots (Matasgordas an Reserva) and for the three studied levels (presence of feces, presence of seeds and number of seeds) were pretty similar to the models obtained when the variable Volume was used. So, we provide the new model in the appendixes and kept the original one in the main text. We also carried out the analysis to study the distribution and level of aggregation of C. humilis. Our cluster analyses showed that Reserva has a greater percentage of isolated C. humilis, which is consistent with our results showing that birds prefer isolated C. humilis for perching in Reserva.

Line 63: “template” should be “temperate”

Thank you. Done.

Line 99: The use of the term “beneficiary” requires some type of description/definition or a reference source.

To address your comment have made the following change at Line 79-81: “Interestingly, in some cases, the same shrub species acting as perches also act as nurse plants, i.e. facilitating the emergence, growth and survival of other plant species, [23] which are designated as “beneficiary species”, promoting the natural (re)colonization [24, 25].” 

Also, at Line 220-222, we now state: “Beneficiary species were woody plants that grow under dwarf palms and emerge from the top of their surface. These species often benefit from C. humilis due to microclimatic improvement and protection against herbivory [42].”

What characteristics define this group and is it context-dependent?

Thank you for such an interesting point. In Garrote et al (2019), some of us proved that there is a consistent and strong positive spatial association between C. humilis and several woody species. This result suggested that the interaction is constant rather than context dependent. Most recently (Garrote et al. 2022), however, we have showed through field experiments that the strength and even the sign of such interaction changes mostly with plant stages (seed survival, seedling emergence and survival), which would suggest context-dependency. However, the overall effect C. humilis on woody species was always either positive or neutral and whenever we detected neutral effects that result seemed associated to limited sample sizes. So, most compelling evidence indicate that, in overall, C. humilis exerts a consistent positive effect on woody species.

Garrote PJ, AR Castilla, JM Fedriani. 2019. The endemic Mediterranean dwarf palm boosts old-field recolonization: implications for restoration. Journal of Environmental Management 250: 109478.

Garrote PJ, A Castilla, JM Fedriani. 2022. Coping with changing plant-plant interactions in restoration ecology: effect of species, site, and individual variation. Applied Vegetation Sciences. DOI: 10.1111/avsc.12644

Line 232: Why 199 simulations? It seems like a rather arbitrary number. Is this based on some initial modeling or preliminary results?

The 199 simulations are the number recommended by Wiegand & Moloney (2013). They estimate the curve asymptotes for each spatial statistic (rank, P-value, etc.). 

Wiegand, T., & Moloney, K. A. (2013). Handbook of spatial point-pattern analysis in ecology. CRC press.

Line 255: “that” should be “than”

Thank you. Done.

 

Journal Requirements:

Reviewed and ensured.

We have added at Lines 131-132: “All permits necessary were granted by the National Park Service (ref. 2019/10) and the Junta de Andalucía (ref. 2019107300002261/IRM/MDCG/mes).”

Thank you. We have now specified JM Fedriani (who is affiliated at CSIC) as corresponding author in the Plos One manuscript submission platform. Also, in the manuscript both JM Fedriani and V Gonzalez-Garcia appear as corresponding authors.

https://figshare.com/articles/dataset/PerchigEffectDwarfPalm_xlsx/19642446

5. We note that Figures 1 and 4 in your submission contain map images which may be copyrighted. All PLOS content is published under the Creative Commons Attribution License (CC BY 4.0), which means that the manuscript, images, and Supporting Information files will be freely available online, and any third party is permitted to access, download, copy, distribute, and use these materials in any way, even commercially, with proper attribution. For these reasons, we cannot publish previously copyrighted maps or satellite images created using proprietary data, such as Google software (Google Maps, Street View, and Earth). For more information, see our copyright guidelines: http://journals.plos.org/plosone/s/licenses-and-copyright.

Sorry, we are not sure to fully understand your concern. Figure 1 and Figure 4 represent study areas sketch maps and study plots, respectively. They have been made by us specifically for this manuscript. To this aim, we used QGIS v3.18 software. So, we don’t see any copyright issue. However, we now provide the photo credit in Figure 2 and Figure 3 headings (Photo credit: Pedro J. Garrote) which was missing in the earlier manuscript version.

 

2. Changes made to address reviewer #1 comments:

Although there is a lot of consideration on the use of artificial perches in restoration, the role of natural perches deserves more investigation, especially in a poorly studied biome under this perspective, as is the case with Mediterranean Europe. In this regard, the authors provide an important dataset based on field observations and fine spatial analyses, which represents a great contribution to the restoration of that particular habitat.

Thank you very much for your suggestions and comments to improve our manuscript.

Despite their efforts to explain the different patterns found between the two study sites (Mastasgordas and Reserva), I think this attempt was a bit confused, perhaps because there may be underlying causes not fully investigated by the authors. I would put this question in the perspective of the field data and not in the analysis itself. One underlying problem may be the unbalanced frequency of birds and seeds in the two areas. From the nine bird species recorded, six are reputed insectivores and just three can be considered omnivores that can take significant amounts of fruit in their diets (https://esajournals.onlinelibrary.wiley.com/doi/abs/10.1890/13-1917.1). These three omnivore species performed more than 50% of the visits recorded in the two areas and probably deposited more seeds per visit beneath the palms than the other ones. Special attention should be paid to the foraging and movement patterns of these three omnivores, since they may account for the general pattern of the seed rain and, perhaps, provide sound explanations for these patterns.

Lines 190-193: “Overall, nine bird species were recorded perching: Cisticola juncidis, Lanius senator, Phoenicurus ochruros, Phylloscopus collybita, Pica pica, Saxicola rubicola, Sturnus unicolor, Sylvia atricapilla and Sylvia melanocephala”. 

All species but Cisticola, Lanius, Pica and Sturnus are reported by Herrera (1984) as frugivorous. However, Pica pica does include fleshy-fruits (Green et al. 2019) and acorns (Martínez-Baroja 2019) in their diet. Additionaly, genus Sturnus has been reported as fruit-eater too (Jordano 1987, Gonzalez-Varo et al. 2017), as well as genus Phoenicurus, Sylvia and Saxicola (Campo-Celada et al. 2022, Carnicer et al. 2009). Therefore, just two of our reported species (Cisticola juncidis and Lanius senator) would not participate directly in seed dispersal. 

That is a good point even though during the fall and early winter (when our study was carried out) most of the recorded bird species (Table 1) shift their diet, at least partially, towards a frugivorous diet. To address your concern, we have added the following text in the discussion (Line 562-570): “However, we sampled during the seed dispersal season, time when some birds change their diet into a more fruit-based one. For instance, it has been reported that P. ochruros, S. atricapilla and S. malenocephala are able to quickly change their diet when fleshy fruits become available, while other species such as S. rubicola show a more gradual change to a frugivorous diet [67, 68]. Also, S. unicolor, P. ochruros, P. collybita, P. pica, and S. rubicola has been reported as fruit-eater, although in lesser quantities [69, 70, 71, 72, 73]. Therefore, although not all recorded bird species contribute equally to seed dispersal, fleshy fruits represent a considerable fraction of most of their diets during the seed dispersal season.”

Campo-Celada M, Jordano P, Benítez-López A, Gutiérrez-Expósito C, Rabadán-González J, Mendoza I. 2022. Assessing short and long-term variations in diversity, timing and body condition of frugivorous birds. Oikos, 2: e08387

Carnicer J, Jordano P, Melían CJ. 2009. The temporal dynamics of resource use by frugivorous birds: a network approach. Ecology, 90(7): 1958-1970.

Gonzalez-Varo JP, Carvalho CS, Arroyo JM, Jordano P. 2017. Unravelling seed dispersal through fragmented landscapes: Frugivore species operate unevenly as mobile links. Molecular Ecology, 26(16): 4309-4321.

Green AJ, Elmberg J, Lovas-Kiss A. 2019. Beyond Scatter-Hoarding and Frugivory: European Corvids as Overlooked Vectors for a Broad Range of Plants. Front. Ecol. Evol., 18.

Herrera CM. 1987. A study of avian frugivores, bird-dispersed plants, and their interaction in Mediterranean scrublands. Ecological Monographs, 54(1): 1-23

Martínez-Baroja L, Pérez-Camacho L, Villar-Salvador P, Rebollo S, Quiles P, Gómez-Sánchez D, Molina-Morales M, Leverkus AB, Castro J, Rey-Benayas JM. 2019. Massive and effective acorn dispersal into agroforestry systems by an overlooked vector, the Eurasian magpie (Pica pica). Ecosphere, 10(12): e02989.

Another point addresses the plant composition and fruit types between the two areas. Rubus ulmifolius is the only recorded fruit consumed that is a functional berry, containing dozens of seeds. The other plant species are single-seeded drupes. This species was not related as common or abundant in Matasgordas, whose seed rain was nil, but its presence was highlighted in Reserva, whose seed rain under C. humilis accounted for 85-90% of the total seed rain. Even a few feces with dozen of R. ulmifolius seeds would certainly bias the results and the comparison between areas when the response variable “number of seeds” is considered. Would the spatial patterns be the same between the areas if R. ulmifolius is excluded from the analysis? If so, the conclusions presented by the authors should be revisited and adjusted to the bias introduced by this species.

I see these points do not invalidate the importance of C. humilis as perches and facilitative species in that habitat, but at least would provide more explanatory power for the distinct spatial patterns found between the two study sites. 

Thank you for your suggestion. We repeated our spatial analysis for the presence of seeds and for the number of seeds excluding Rubus’ seeds. The results obtained are pretty similar to the ones obtained initially when we included those seeds in the analysis. The results from such a new analysis are now included as Supporting Information (S2 Spatial seed analysis excluding Rubus’ seeds).

Besides that, I would recommend the authors to improve the English text, as some sentences are somewhat difficult to understand.

Thank you. We have carefully revised the manuscript several times to improve it language use and clarity. The revised manuscript has considerably improved in this respect.

Bellow, I add some other specific points to the authors.

Page 4, lines 94-96: To question whether there are differences in the role of C. humilis as perches between the two areas presupposes some previous knowledge of the distributional pattern of this species between the areas. Although the authors provide a detailed description of the study areas in the proper session (pages 6 and 7) it should be enlightening to give a hint here on what triggered the idea that the plant role could be different between areas. Certainly, these two areas were not chosen at random.

Garrote et al. 2019 documented that Reserva presents higher density of shrubs than our study plot at Matasgordas. Such a difference in shrub cover led to a greater abundance of fleshy fruits in Reserva which likely makes this site more attractive to frugivorous birds.

Lines 137-138: “Study areas are different in terms of vegetation and human-use history [4, 38]”.

Lines 147-152: “Due to such past human activity, Matasgordas is now formed, mainly, by two habitats: (i) Scrubland dominated by P. lentiscus shrubs with some Q. suber and O. europaea var. sylvestris trees [4, 33] and (ii) an old-field, where our study plot was set, which woody vegetation is mainly composed by animal-dispersed native plants such as C. humilis, P. lentiscus, D. gnidium and P. bourgaeana, Asparagus aphyllus L., Halimium halimifolium (L.) Willk and Cistus salviifolius L. and scattered Q. suber and O. europaea var. sylvestris trees [4]”.

Lines 167-175: “Reserva (…) has been historically managed by human for agriculture, hunting, cattle grazing and tree felling, especially O. europaea var. sylvestris and Q. suber [41]. Reserva was protected earlier than Matasgordas, in 1964 and it has been recovering ever since, being recolonised by animal-dispersed plants such as A. aphyllus, C. humilis, Phillyrea angustifolia L. or R. ulmifolius [4]. This scrubland is dominated by H. halimifolium and Stauracanthus genistoides (Brot.) Samp. with scattered trees of Q. suber, O. europaea var. sylvestris and Pinus pinea [33]. The study plot at Reserva presents higher density of shrubs than the one at Matasgordas. Such a difference in shrub cover led to a greater abundance of fleshy fruits in Reserva, which likely makes this site more attractive to frugivorous birds [4].”.

Page 6, line 142: Delete the period after bourgaeana.

Thank you. Done.

Page 7, lines 177-178: What was the total sampling effort for bird perching for the separate areas?

Thank you for your comment. To address it, we have added the following text (Line 189-190) in the revised manuscript: “During the sampling, which lasted about 3h per day in Matasgordas and about 2h in Reserva, any observed bird perching on focal C. humilis individuals was recorded.”

Page 9, lines 189-192: What do you mean by “palm interior”? Does it include the foliage surface in the interior and also the soil under the foliage? Was the open area of a similar surface associated with the palms bare soil? 

Palm interior refers to the foliage surface since the soil under the palm was unreachable in most of the cases. Thus, we have replaced in the manuscript “palm interior” for “palm foliage surface”.

Yes, the open area associated to each palm was of similar surface (than the associated palm) and mostly in bare soil with scarce vegetation, mainly grasses. This open area was 3-4m from the associated C. humilis.

Lines 200-204: “Every target C. humilis was checked for bird feces and regurgitations by carefully checking the palm foliage surface. We also evaluated whether seeds arrive at open spaces without palms. To this aim, we checked for the presence of bird feces and regurgitations in an open area associated to each focal C. humilis. These areas were of similar surface and mostly located in bare soils with scarce vegetation (mainly grasses) and 3-4m from the associated C. humilis”.

What were the criteria used to distinguish between bird-and-mammal dispersed seeds in these areas?

Most sampled seeds were found within bird feces which are easily distinguishable from mammal feces based on size, shape, color and smell. For example, the uric acid in bird feces forms a conspicuous and white sticky paste that makes them easily to identify. Also, the diameter of feces of target birds (a few millimeters) was much smaller than for feces of local frugivorous mammal species (several centimeters). 

Page 14, lines 312-313: Since the number of feces and number of seeds are two interrelated response variables, what is the weight of the regurgitated seeds in the analysis? How much regurgitated seeds contributed to the total number of seeds sampled?

Regurgitated seeds were just a small fraction of the total sample. Their number was too small to really contribute to the final analysis.

Page 16, line 375: Consider this writing: “Otherwise, in Reserva, the probability …”

Thank you. We have changed the sentence. Line 395-397 (previously line 375): “Results in Reserva substantially differed from results from Matasgordas. The probability of two C. humilis separated by distance r, having both seeds (p11(r) statistic), was similar to expected of random labelling for all spatial scales (Fig 6B).”

Page 2, line 501: Insert “of” after “chances”.

Thank you. Done.

 

3. Changes incorporated to address reviewer #2 comments: 

Unmasking the perching effect of the pioneer Mediterranean palm Chamaerops humilis L.

I have now carefully read the manuscript. I found the authors approach very interesting using SPPA to unravel the perching effect. Using this spatial analysis they evaluate several spatial patterns of “perching” effect and whether the seed arrival via frugivorous birds is related to the spatial distribution of the C. humilis in two disturbed scenarios. I think the manuscript could be interesting will be interesting for readers interested in restoration or ecological succession drivers.

Thank you very much for your suggestions and comment to improve our manuscript.

I think the manuscript needs to improve the readability mostly in the methods and the results section (see comments below). 

We have considerably improved the clarity and readability of the whole manuscript, including the methods and results sections.

I have concerns with the general work. On one hand, I think that it needs more information about the state of the selected plots. For instance, the main objective is to evaluate the perching effect based on the plants distribution, however, the authors did not show the spatial distribution of the individual plants, plants density, size of the plot, etc. that in my opinion has to be in the article. 

Thank you for communicating this concern. Both Figure 1 and Figure 4 show the spatial distribution of the individual C. humilis. Following your suggestion, plant densities and plot sizes are now included in the Material and Methods section:

Line 136-137: “We selected two study plots 10 km apart called Matasgordas (area sampled: 13.9 ha) and Doñana Biological Reserve (area sampled: 21.4 ha)”.

Line 182-184: “The density of C. humilis plants in Matasgordas’ plot (7.78 individuals/ha) was pretty similar than in Reserva’ plot (8.41 individuals/ha).”.

On the other hand, your main conclusions are based on some patterns found in one plot (male plants, isolation). This is my major concern because when you present the analysis of the other plot the main patterns did not applied to other ecological context. It seems that the results only applied under certain circumstances. I think this is a very common problem when studying the effects of perching which makes it very difficult to generalize the drivers of this process.

We agree in that is very difficult to generalize the drivers of the perching effect. Differences in results between our study areas likely relate to the fact that they present a different state of ecological succession, an issue that is detailed in our Discussion and could be used by managers to decide, for example, whether particular areas are adequate to promote seed arrival using natural or artificial perches.

Abstract

L39- “dispersed seeds We detected” missed a point

Thank you. Done.

Introduction

L97: Do exist some individuals of C. humilis that act as “hotspots” of dispersed seeds? Change for: Do any C. humilis individuals act as "hot spots" for seed dispersal?

Changed for: Do particular C. humilis individuals act as "hot spots" of seed arrival, and if so, why? (Lines 99-100, previously Line 97).

Methods

L134-144. With this explanation it is unclear where did you work.

Reading some works made in the area it seems that Matasgordas has a higher plant density than Reserva. For instance, In the article by Fedriani and Delibes (2011) when the description of the Matasgordas site is made, it seems to have a higher density of individuals, same situation in the article by Jácome et al. 2016 where the spatial patterns of this species are studied where they also report higher number of individuals in the area. Thus, it would be interesting to specify that you do it “in a low density area in order to be comparable with Reserva.”

Thank you for your comment. The density for C. humilis is highly spatially heterogeneous both in Matasgordas and in Reserva areas. Thus, it is expectable that density changes depending on the specific location of the study plots in both areas. In our selected study plots, the density of C. humilis plants in Matasgordas (7.78 individuals/ha) was pretty similar to that in Reserva (8.41 individuals/ha) but, again, this could change if the plots were set in different specific locations. Ultimately, Matasgordas and Reserva were not too different in terms of palm density.

Our previous studies (Fedriani and Delibes 2011, Jácome et al. 2016) were carried out in Matasgordas and Reserva but their study plots were not the same that the pots of the present study; so, no wonder they reported other C. humilis densities. For instance, the study by Fedriani and Delibes (2011) was carried out in a patch of Mediterranean scrubland within Matasgordas area (where C. humilis density is certainly very high), whereas our present study was conducted in the old-field within Matasgordas area (where C. humilis density is much lower than in the scrubland). 

To prevent possible misunderstandings in relation to this issue, we now consistently use throughout our manuscript the term study plot instead of study site. 

L163- Related with my previous comment. Although Reserva was protected earlier authors have to consider that there still traditional uses in the area, thus plants in the Reserva are submitted to a high herbivorous pressure due to the presence of horses, cattle and high density of cervids. This pressure has reflected in the lower density of individuals explained above.

As mentioned above, the densities of C. humilis in both Matasgordas (7.78 ind. ha-1) and Reserva (8.41 ind. ha-1) were not too different but, they could change depending on where the study plots are located.

Suggested papers: 

-Fedriani, J. M., & Delibes, M. (2011). Dangerous liaisons disperse the Mediterranean dwarf palm: Fleshy- pulp defensive role against seed predators. Ecology, 92, 304–315.

-Jácome-Flores, M. E., Delibes, M., Wiegand, T. & Fedriani, J. M. Spatial patterns of an endemic Mediterranean palm recolonizing old fields. Ecol. Evol. 6, 8556–8568 (2016).

L170-Every target C. humilis individual (109 and 180 in Matasgordas and Reserva, respectively)

Thank you. Done.

SPPA requires the delimitation of an area in which all target individuals of the study species are georeferenced. I suppose this was the case, however it is not clear in the methodology. I suggest to put more information such as density and/or no. of individuals, area sampled, etc. for each selected plot. Also this information will clarify more the differences that you found between plots.

Thank you. The study plots are shown in Fig. 1. We have added the suggested information: 

Lines 136-137: “We selected two study plots 10 km apart called Matasgordas (area sampled: 13.9 ha) and Doñana Biological Reserve (area sampled: 21.4 ha)”.

Lines 181-184: “Every target C. humilis (109 and 180 in Matasgordas and Reserva, respectively) was individually georeferenced with a submetric GPS Leica 1200. The density of C. humilis plants in Matasgordas’ plot (7.78 individuals/ha) was pretty similar than in Reserva’ plot (8.41 individuals/ha)”. 

Discussion

L463-L478- Reduce the length of this paragraph, there is too much mention of dispersal mechanisms of other species. This section should only focus on the ability of C. humilis to attract different dispersers including mammals.

We changed the paragraph to highlight the C. humilis ability to attract seed dispersers as you suggest.

Lines 500-509: “Besides, it is apparent that some seeds can emerge seedling beneath C. humilis, since all seeds collected belonged to beneficiary woody species found within C. humilis. Nevertheless, we did not find seeds of the beneficiary species C. salviifolius, P. bourgaeana, Q. suber and S. genistoides, which are usually associated to C. humilis. Most of these plant species are seldom dispersed by birds. Other animals are more likely to disperse them, such as insects for C. salviifolius or S. genistoides seeds [56, 57, 58] or mammals, such as ungulates (red deer, wild boars, domestic cattle) and carnivores (Eurasias badger, red fox) for P. bourgaeana seeds [59]. Therefore, the presence of these woody species associated to C. humilis reveals that it does not attract just bird-dispersed plants but also mammal-dispersed plants, accentuating its role in habitat restoration.”

L484-485 I suggest to the authors to do SPPA to the individual plants to unravel the spatial patterns of the “Matasgordas and Reserva C. humilis populations”, this analysis could be useful to explain some of the patterns. For instance maybe the patterns of the plants in Matasgordas tend to be more random explaining your findings without any over explanation. On the other hand maybe Reserva population its characterized by more isolated plants that had enough height to function as a perch for birds.

Thanks for your suggestion. We carried out some cluster analysis in order to quantify the C. humilis distribution and level of aggregation. To evaluate the fit of the observed point patterns in both Matasgordas and Reserva to different cluster processes, four different summary functions were used: pair correlation function g(r), L-function L(r), Spherical contact distribution HS(r) and nearest neighbor distribution function D(r).

In both study plots, the four summary functions showed good fits with the selected null models, being the model-predicted values mostly confined within the simulation envelopes. The distribution of C. humilis in Matasgordas and Reserva were both best described by a double-clustered component pattern with a random component pattern, being Reserva the one with a greater proportion of isolated C. humilis individuals. This may relate to the observed tendency found in Reserva of bird’s preference for isolated C. humilis, which is now mentioned in the Discussion.

We added the results of cluster analyses as Supporting Information (S1 Cluster Analysis) and are quoted in the description of the study plots.

L528-529: …while the surface of Chamaerops humilis was found significant in Reserva

I suggest to change surface to cover area.

Thank you for your suggestion. We have removed that line in the revised version.

L526-534. Sometimes is difficult and confusing to follow the explanations or results because sometimes appear first Matasgordas and then Reserva. I suggest to the authors to be consistent with the appearance order through all document.

Thank you. The order is now consistent, as suggested.

L550 Change: What is more to Furthermore or Additionally

Thank you. Done.

L552- Unexpectedly, it was found that larger C. humilis (i.e. those with larger volume) were less suitable for seed arrival, at least in Reserva. -Change “it was found” to “we found”

Thank you for your suggestion. We have removed that line in the revised version.

-I suggest to the authors not use volume but instead the interaction between cover area (surface) and height. This interaction could reveal some perching preferences regarding this two variables. The variety of shapes in this palm make it difficult to standardize the volume as a “common variable” between all individuals.

Although this species can be highly variable in shape, we found that C. humilis in our study plots were very similar in shape, with a consistent dome shape. This uniformity made us to choose the variable ‘volume’. 

However, following your suggestion, we carried out the GLM with the interaction between Height (m) and Area (m2) instead of the variable Volume (m3) and the results obtained were pretty similar to the ones obtained in the initial GLM. Results from this new analysis are shown in the Supporting Information (S3 Effect of palm traits).

L554-We speculate that male C. humilis could be selected over females because they lack fruits. C. humilis fruits are too big for birds, but they are often eaten by medium-sized carnivores such as the European badger and the red fox [33]. The presence of these predators could dissuade frugivorous birds from visit female C. humilis in fruit, and thus perching more often male palms.

I suggest to put some information about the abundance of “potential predators” in each study area to reinforced this idea. Another explanation to this pattern (in my opinion more likely) is the presence of rodents and rabbits beneath the female plants. Rodents and rabbits are also highly attracted to C. humilis fruits, in this foraging process they could predate the seeds dropped in the perching process, limiting the number of seeds. In this context, the same situation could be happening in aggregated C. humilis individuals that are preferred by rodents due to high vegetation cover.

Thank you for your comment. On the one hand, the presence and abundance of potential predators in this area, such as red foxes and badgers, have been studied earlier (Fedriani et al. 1998, Fedriani et al. 2008). On the other hand, all seeds collected were found at a certain height at the palm leaves, being pretty much inaccessible for those mammals (especially rabbits). Further, during our intensive monitoring of C. humilis using camera traps we have not recorded ever rodent foraging on leaves at the top surface of this palm, where seeds dispersed by birds were collected. 

L564- It seems that your results could be useful for restoration planification where the C. humilis is a very useful species. However the inconsistency of the results between areas and the unclear relation between the traits masked by the context dependency seems to be an issue to the applicability of the study. For instance the efficiency of the perching effect seems to be related to male plants, if we use this results as it comes we would be deprecated female plants that would be useful to attract super-efficient fleshy fruit dispersers such as badgers or foxes.

Thank you for your comment. We have included the following sentences in the revised manuscript (Lines 593-597): “The use of C. humilis in ecological restoration should follow a balance between male and female plants. Male plants would attract preferably birds that would disperse seeds of woody species present in the surrounding area, while female plants would attract medium-sized carnivores and ungulates that would mostly disperse seed of species such as P. bourgaena and C. humilis.”

---

## [Decision Letter · Decision Letter 1]

29 Jun 2022

PONE-D-22-01669R1Unmasking the perching effect of the pioneer Mediterranean palm *Chamaerops humilis* L.PLOS ONE

Dear Dr. Fedriani,

Thank you for submitting your manuscript to PLOS ONE. After careful consideration, we feel that it has merit but does not fully meet PLOS ONE’s publication criteria as it currently stands. Therefore, we invite you to submit a revised version of the manuscript that addresses the points raised during the review process.

I appreciate the authors’ thoroughness and thoughtfulness in addressing the numerous comments and suggestions by the original reviewers. The paper examines an interesting restoration question about the role of perch plants for seed dispersal in disturbed environments.  While it makes an important and timely contribution to our understanding of the effects of structure on seed dispersal the manuscript needed increased detail, clarity and improvement in the description of the study design, the analysis and the interpretation of the results.  The revised version is much clearer and the results are much better supported.  The new analyses are illuminating and provide important detail to the results.  As the reviewers suggest there are still a few minor revisions that could further strengthen the paper and improve the readability.  Please note, though, the reviewers for this version are not the original reviewers so they have provided a fresh perspective on the revised manuscript.  As Reviewers 3 and 4 highlight, there are a number of places where some additional editing (e.g., improving the grammar) will improve the readability.  As reviewer 3 suggests, it would also be help to refer to the sites in a consistent order to aid the reader in tracking the differences.  Both reviewers have included detailed suggestions and clarifying language that should be considered throughout, which are included in the annotated versions.  In addition, they have provided questions highlighting places where additional information or a rephrasing may be necessary.  With these minor revisions, the paper should be well crafted to contribute to our understanding of the ecological implications of structure in restoration on seed dispersal.

We look forward to receiving your revised manuscript.

Kind regards,

Karen Root, Ph.D.

Academic Editor

PLOS ONE

Journal Requirements:

Reviewers' comments:

Reviewer's Responses to Questions

**Comments to the Author**

1. If the authors have adequately addressed your comments raised in a previous round of review and you feel that this manuscript is now acceptable for publication, you may indicate that here to bypass the “Comments to the Author” section, enter your conflict of interest statement in the “Confidential to Editor” section, and submit your "Accept" recommendation.

Reviewer #3: (No Response)

Reviewer #4: (No Response)

2. Is the manuscript technically sound, and do the data support the conclusions?

Reviewer #3: Yes

Reviewer #4: Yes

3. Has the statistical analysis been performed appropriately and rigorously? 

Reviewer #3: Yes

Reviewer #4: I Don't Know

4. Have the authors made all data underlying the findings in their manuscript fully available?

Reviewer #3: Yes

Reviewer #4: No

5. Is the manuscript presented in an intelligible fashion and written in standard English?

Reviewer #3: Yes

Reviewer #4: Yes

6. Review Comments to the Author

Reviewer #3: A note that I was not a reviewer on the first round of the manuscript. I thought for the most part that the authors did a thorough job of responding to prior reviewers’ comments. Generally, the paper is clearly written and the analyses seemed sound. I did, however, make quite a few edits directly on the manuscript and think the manuscript needs another round of revisions. Here I just summarize a few key points.

1. Most of my comments are grammatical corrections (using either comments or in red text). I started doing a thorough edit but that is challenging with a pdf so only did so on the first couple of pages. The paper needs a thorough edit by somebody with a strong command of English grammar.

2. The authors refer to their two sites by the actual site names, but those site names are meaningless to the reader. Then the authors have to reminder the different characteristics of the sites. I suggest that the authors name the sites for the paper using what they think is the most distinguishing feature – e.g. early-successional and late-successional site.

3. The authors overstep their data in their restoration recommendations. They have data from only two sites and the results differ in the two sites, which is typical of restoration – results are often site-specific. So they have a limited ability to generalize. They also make recommendations in that section that don’t stem from the data presented. I strongly recommend that they combine their “recommendations” and “conclusions” section into one section and condense the restoration conclusions to 1 paragraph.

4. The authors frequently put modifying clauses in parentheses. In most cases, these should be part of the main sentence.

Please see the manuscript for additional comments and edits.

Reviewer #4: I made a series of minor suggestions/comments directly on the ms.

Regarding the availability of data, I found a vague statement that "all data are fully available without restriction", without any indication of how to access it.

7. PLOS authors have the option to publish the peer review history of their article (what does this mean?). If published, this will include your full peer review and any attached files.

Reviewer #3: **Yes: **Karen Holl

Reviewer #4: No

---

## [Author Response · Author response to Decision Letter 1]

11 Jul 2022

RESPONSE TO EDITOR COMMENTS

I appreciate the authors’ thoroughness and thoughtfulness in addressing the numerous comments and suggestions by the original reviewers. The paper examines an interesting restoration question about the role of perch plants for seed dispersal in disturbed environments. While it makes an important and timely contribution to our understanding of the effects of structure on seed dispersal the manuscript needed increased detail, clarity and improvement in the description of the study design, the analysis and the interpretation of the results. The revised version is much clearer and the results are much better supported. The new analyses are illuminating and provide important detail to the results. As the reviewers suggest there are still a few minor revisions that could further strengthen the paper and improve the readability. Please note, though, the reviewers for this version are not the original reviewers so they have provided a fresh perspective on the revised manuscript. As Reviewers 3 and 4 highlight, there are a number of places where some additional editing (e.g., improving the grammar) will improve the readability. As reviewer 3 suggests, it would also be help to refer to the sites in a consistent order to aid the reader in tracking the differences. Both reviewers have included detailed suggestions and clarifying language that should be considered throughout, which are included in the annotated versions. In addition, they have provided questions highlighting places where additional information or a rephrasing may be necessary. With these minor revisions, the paper should be well crafted to contribute to our understanding of the ecological implications of structure in restoration on seed dispersal.

Thank you very much for your positive comments. We have carefully followed comments by reviewer 3 and 4 to improve further the grammar and readability of our manuscript. We now refer to the study sites in a consistent order to aid the reader in tracking the differences. Also, we have included additional information or a rephrasing when necessary.

RESPONSE TO REVIEWER 3 COMMENTS

Reviewer #3: A note that I was not a reviewer on the first round of the manuscript. I thought for the most part that the authors did a thorough job of responding to prior reviewers’ comments. Generally, the paper is clearly written and the analyses seemed sound. I did, however, make quite a few edits directly on the manuscript and think the manuscript needs another round of revisions. Here I just summarize a few key points.

Thank you very much for your helpful comments.

1. Most of my comments are grammatical corrections (using either comments or in red text). I started doing a thorough edit but that is challenging with a pdf so only did so on the first couple of pages. The paper needs a thorough edit by somebody with a strong command of English grammar.

We have addressed all grammatical corrections pointed by both reviewers. We have also revised again the whole manuscript ton improve its clarity and flow of logic.

2. The authors refer to their two sites by the actual site names, but those site names are meaningless to the reader. Then the authors have to reminder the different characteristics of the sites. I suggest that the authors name the sites for the paper using what they think is the most distinguishing feature – e.g. early-successional and late-successional site.

The names of both study sites have been changes as suggested throughout the manuscript.

3. The authors overstep their data in their restoration recommendations. They have data from only two sites and the results differ in the two sites, which is typical of restoration – results are often site-specific. So they have a limited ability to generalize. They also make recommendations in that section that don’t stem from the data presented. I strongly recommend that they combine their “recommendations” and “conclusions” section into one section and condense the restoration conclusions to 1 paragraph.

All these suggestions have been included in the revised manuscript.

4. The authors frequently put modifying clauses in parentheses. In most cases, these should be part of the main sentence.

We have included in the main sentences several clauses that used to be in parentheses. 

Please see the manuscript for additional comments and edits.

We have addressed all comments and edits provided in the annotated manuscript. Below, I summarize the most relevant ones:

L472. How can height be negatively relate to number of seeds and size positively related?

There was a mistake in this sentence. Both height and area negatively relate to number of seeds, whereas volume was positively related. We have corrected these sentences. 

L517-518. Or is it that the feces were more concentrated in single plants because there was only one?

Indeed, in late successional study plot (Reserva) there were more individuals that receive seeds (31) than in the early successional study plot (Matasgordas; 20).

RESPONSE TO REVIEWER 4 COMMENTS

Reviewer #4: I made a series of minor suggestions/comments directly on the ms.

Thank you for your comments and suggestions. All of them have been addressed in the revised manuscript.

Regarding the availability of data, I found a vague statement that "all data are fully available without restriction", without any indication of how to access it.

The data used for this study is fully available at:

https://figshare.com/articles/dataset/PerchigEffectDwarfPalm_xlsx/19642446

---

## [Decision Letter · Decision Letter 2]

8 Aug 2022

Unmasking the perching effect of the pioneer Mediterranean dwarf palm *Chamaerops humilis* L.

PONE-D-22-01669R2

Dear Dr. Fedriani,

We’re pleased to inform you that your manuscript has been judged scientifically suitable for publication and will be formally accepted for publication once it meets all outstanding technical requirements.

Kind regards,

Ignasi Torre

Academic Editor

PLOS ONE

Additional Editor Comments (optional):

The article was reviewed again by rev. 3 and 4 and minor changes need to be done (some comments not addressed, grammatical and typographical errors).

Reviewers' comments:

Reviewer's Responses to Questions

**Comments to the Author**

1. If the authors have adequately addressed your comments raised in a previous round of review and you feel that this manuscript is now acceptable for publication, you may indicate that here to bypass the “Comments to the Author” section, enter your conflict of interest statement in the “Confidential to Editor” section, and submit your "Accept" recommendation.

Reviewer #3: All comments have been addressed

Reviewer #4: All comments have been addressed

2. Is the manuscript technically sound, and do the data support the conclusions?

Reviewer #3: Yes

Reviewer #4: Yes

3. Has the statistical analysis been performed appropriately and rigorously? 

Reviewer #3: Yes

Reviewer #4: I Don't Know

4. Have the authors made all data underlying the findings in their manuscript fully available?

Reviewer #3: Yes

Reviewer #4: Yes

5. Is the manuscript presented in an intelligible fashion and written in standard English?

Reviewer #3: No

Reviewer #4: Yes

6. Review Comments to the Author

Reviewer #3: The authors have addressed all my substantive comments but there remain a number of typographical and grammatical errors. That said it would take extensive time for me to make those grammatical edits on a pdf version of the document and I don't have access to the Word version. I am not sure what type of resources PLOS One provide for English language proofing but the paper needs a careful edit for grammar.

Also, there should be a space between the numbers and units (e.g. 6 m not 6m) throughout. If the measurement is used as a an adjective then a dash is included (e.g. 6-m transect).

Reviewer #4: Two comments I raised before have not been addressed in this revision (pages 9 and 10 of the ms). I left them marked in the ms PDF file attached.

7. PLOS authors have the option to publish the peer review history of their article (what does this mean?). If published, this will include your full peer review and any attached files.

Reviewer #3: No

Reviewer #4: No

---

## [Editor Report · Acceptance letter]

12 Aug 2022

PONE-D-22-01669R2 

Unmasking the perching effect of the pioneer Mediterranean dwarf palm *Chamaerops humilis* L. 

Dear Dr. Fedriani:

I'm pleased to inform you that your manuscript has been deemed suitable for publication in PLOS ONE. Congratulations! Your manuscript is now with our production department. 

Kind regards, 

on behalf of

Dr. Ignasi Torre 

Academic Editor

PLOS ONE